# A Unified Sparse Attention via Multi-Granularity Compression

**Siran Liu** [1 2]  **Zane Cao** [2]  **Yongchao He** [2]

## Abstract

Efficient long-context understanding is increasingly vital for large language model (LLM) applications such as multi-turn dialogue and program analysis. However, the core *self-attention* scales quadratically with sequence length, creating a fundamental computational bottleneck. Existing sparse attention methods alleviate this issue but face trade-offs: training-based methods are costly and cannot be directly applied as acceleration plugins for other models, while inference-time methods often compromise efficiency or cross-modal generality. To address these limitations, we present UniSparse, a unified mechanism that introduces the notion of *composite tokens*—compact representations that aggregate multi-granularity contextual information. Building on this abstraction, UniSparse dynamically constructs sparse attention through *multi-granularity compression* and *block-level selection*, enabling efficient and hardware-friendly execution on GPU. Across multiple modalities and tasks ranging from synthetic benchmarks to real-world applications, UniSparse consistently surpasses state-of-the-art sparse attention methods (e.g., MInference, XAttention, Flex-Prefill) in both accuracy and efficiency, achieving $\geq$ 99% of full-attention accuracy and up to $2.61\times$ faster attention computation than FlashAttention.

## 1. Introduction

The unprecedented success of Transformer-based (Vaswani et al., 2017) large language models (LLMs) across natural language processing, computer vision, and multi-modal tasks is largely driven by their core *self-attention* mechanism. As applications increasingly demand understanding and generation over long contexts—ranging from multi-turn dialogue and document reasoning to code and program analysis—efficiently processing long sequences has become critical. However, self-attention scales quadratically ($O(L^2)$) with sequence length $L$, becoming the dominant bottleneck for long-context processing. For example, extending context from 4K to 128K tokens—a trend in recent models such as DeepSeek (DeepSeek-V3.2-Exp)—increases attention cost by about $1024\times$, amounting to trillions of floating-point operations. While tensor, pipeline, or sequence parallelism (He et al., 2025a; Kim et al., 2025) can distribute memory across devices, they cannot mitigate the quadratic compute growth or the substantial inter-GPU communication it incurs.

A variety of methods have been proposed to mitigate this quadratic overhead by introducing sparsity into self-attention, either learned during training or applied post-hoc at inference. Training-based approaches (Yuan et al., 2025; Lu et al., 2025) learn data-dependent sparsity patterns that align well with model representations. In contrast, inference-time methods—including static (Zaheer et al., 2020; Xiao et al., 2023; He et al., 2025b) and dynamic ones (Jiang et al., 2024; Gao et al., 2024; Xu et al., 2025; Lai et al., 2025)—apply sparsification to pre-trained models. To maximize GPU parallelism, these strategies employ a *block-wise* design that restricts computation to selected query–key block pairs indicated by a *sparse mask matrix* $\mathcal{M}$.

However, existing approaches expose a clear system tension between fidelity and efficiency. Training-coupled methods achieve accuracy but lack plug-and-play generality, while inference-time heuristics offer speed at the cost of robustness and adaptability (validated in §5). In practice, this trade-off centers on how $\mathcal{M}$ is determined—typically via a lightweight *proxy computation* that estimates block importance without performing full attention. Existing proxies either rely on expensive attention-like operations or on oversimplified heuristics that fail to capture semantic relevance, leading to a difficult trade-off between proxy accuracy and cost. This persistent gap raises a question: *is there a sparse attention mechanism that is both hardware-efficient and robust, generalizing across modalities without retraining?*

This paper presents UniSparse, a unified, hardware-friendly mechanism for *dynamic sparse attention* that operates at inference time and generalizes across modalities without modality-specific tuning. The key insight is that *semantically meaningful attention patterns can be faithfully inferred*

---

[1]Peking University, Beijing, China [2]ScitiX AI. Correspondence to: Yongchao He <yongchao-he@outlook.com>.

*Proceedings of the $43^{rd}$ International Conference on Machine Learning*, Seoul, South Korea. PMLR 306, 2026. Copyright 2026 by the author(s).

*within a drastically compressed token space*. Instead of evaluating token importance at full resolution, UniSparse constructs *composite tokens*—coarse-grained summaries formed through spatial aggregation—that preserve essential contextual structure while reducing proxy computation cost.

Building on this insight, UniSparse implements a *multi-granularity compression mechanism* that projects query and key representations into a reduced space along both sequence and head dimensions, enabling lightweight yet expressive proxy computations. A *dynamic block selection algorithm* then computes attention scores in compressed space, aggregates them to block-level importance, and identifies the most salient regions. The resulting mask $\mathcal{M}$ guides efficient block-sparse attention using standard kernels such as FlashAttention (Dao et al., 2022), ensuring both software portability and hardware efficiency.

Together, these techniques enable UniSparse to deliver high-fidelity sparse attention without model retraining or task-specific heuristics. Across text and multi-modal benchmarks, UniSparse consistently surpasses state-of-the-art sparse methods in both accuracy and efficiency, using 1.5–2× fewer attention blocks. End-to-end, UniSparse attains up to 2.61× faster attention with over 99% accuracy retention, demonstrating that composite tokens generalize effectively across modalities and architectures.

In summary, this paper makes the following contributions: (1) We characterize the trade-off between attention accuracy and computational cost in existing sparse attention methods, highlighting its system-level implications; (2) We propose UniSparse, which realizes efficient sparse attention by introducing composite tokens for compact multi-granularity context representation, enabled through multi-granularity compression and dynamic block selection; and (3) We implement and evaluate UniSparse across diverse workloads, achieving up to 2.61× speedup in attention computation with ≤1% accuracy loss.

**Conflict of Interest Disclosure.** The authors declare no financial conflicts of interest related to this work.

## 2. Background

### 2.1. LLM Inference

Modern LLMs rely on stacked Transformer blocks combining attention and MLP layers. During inference, the *prefill phase*—a full-sequence forward pass with quadratic attention cost—dominates runtime in long-context tasks. This work targets prefill optimization, while the subsequent *decoding phase* presents orthogonal optimization opportunities.

For a sequence of length $L$ with query, key, and value

matrices $\mathbf{Q}, \mathbf{K}, \mathbf{V} \in \mathbb{R}^{L \times d_k}$, attention computes $\mathbf{O} = \text{softmax}(\mathbf{Q}\mathbf{K}^T/\sqrt{d_k})\mathbf{V}$, where forming the $L \times L$ score matrix dominates with complexity $\mathcal{O}(L^2 d_k)$. Modern GPU implementations like FlashAttention (Dao et al., 2022) partition the sequence into $N = L/S$ blocks and compute attention tile-by-tile, improving memory efficiency for long sequences.

### 2.2. Block-Sparse Attention

While block-wise designs improve GPU throughput, the quadratic complexity $\mathcal{O}(L^2 d_k)$ still dominates for long sequences. Block-sparse attention (Jiang et al., 2024; Guo et al., 2024) addresses this by computing attention only on selected block pairs: a binary mask $\mathcal{M} \in \{0,1\}^{N \times N}$ specifies which query-key block pairs to compute, with $\mathcal{M}_{ij} = 1$ indicating query block $i$ attends to key block $j$. For autoregressive LLMs, causality requires $\mathcal{M}_{ij} = 0, \forall j > i$.

The attention output is computed by masking the attention scores before softmax: $\mathbf{O} = \text{softmax}\left(\frac{\mathbf{Q}\mathbf{K}^T}{\sqrt{d_k}} + \tilde{\mathcal{M}}\right)\mathbf{V}$, where $\tilde{\mathcal{M}}$ is the token-level mask derived from $\mathcal{M}$, with $0$ for selected blocks and $-\infty$ for masked positions.

The computational savings are determined by the sparsity ratio $\rho = 1 - \frac{2}{N(N+1)}\sum_{i \geq j}\mathcal{M}_{ij}$ under causality. The total cost of block-sparse attention can be decomposed as

$$C_{\text{total}} = \underbrace{C_{\text{select}}}_{\text{Mask Determination}} + \underbrace{(1-\rho) \cdot C_{\text{full}}}_{\text{Sparse Computation}} \quad (1)$$

where $(1-\rho) \cdot C_{\text{full}} = (1-\rho) \cdot \mathcal{O}(L^2 d_k)$ is the cost of full attention on selected blocks, and $C_{\text{select}}$ is the overhead of determining $\mathcal{M}$. For a given sparsity $\rho$, most methods have similar sparse computation costs, so $C_{\text{select}}$ becomes the main factor distinguishing their efficiency. The fundamental challenge is *designing a selection mechanism that balances low $C_{select}$ with high accuracy in identifying important blocks.*

## 3. Related Work

To reduce the quadratic cost of attention, numerous methods exploit sparse patterns. These approaches differ along two axes: *when* sparsity is applied—training or inference—and *how* it is defined—static versus dynamic. We review these methods and highlight their trade-offs in adaptivity, overhead, and generality.

### 3.1. Training-Based Sparse Attention

Training-based methods embed sparse structures during pre-training or fine-tuning, such as DeepSeek's hierarchical sparse patterns (Yuan et al., 2025) and Kimi's learned block router (Lu et al., 2025). While representing significant architectural advances, they require model retraining and

thus cannot directly accelerate existing pre-trained models. *Inference-time* sparsification serve as an orthogonal, plug-and-play solution for this broader deployment scenario.

### 3.2. Inference-Time Sparse Attention

#### 3.2.1. INFERENCE-TIME STATIC ATTENTION

Static methods apply pre-defined, input-agnostic sparse patterns during inference, incurring zero selection overhead. Examples include BigBird (Zaheer et al., 2020), which combines sliding window, global, and random attention; StreamLLM (Xiao et al., 2023), which retains early sink tokens; and TriangleMix (He et al., 2025b), which skips intermediate Q–K block interactions in deeper layers. While computationally efficient, fixed patterns lack adaptability to input-dependent long-range dependencies. Originally tuned for text, most designs struggle with spatio-temporal or cross-modal relationships in video or audio.

#### 3.2.2. INFERENCE-TIME DYNAMIC ATTENTION

Dynamic methods construct the mask $\mathcal{M}$ on-the-fly via lightweight proxy computation, balancing proxy accuracy, computational cost, and hardware efficiency. Existing approaches fall into two families.

*(1) High-Fidelity Proxy Methods.* These methods approximate dense attention patterns with expressive but costly proxies. MInference (Jiang et al., 2024) conducts offline searches to assign predefined patterns, while SeerAttention (Gao et al., 2024) introduces learnable gating networks for runtime selection. Although effective, such designs require model-specific pre-processing or additional training, compromising generality and plug-and-play usability.

*(2) Low-Complexity Proxy Methods.* An alternative line employs simple heuristics to minimize runtime overhead. XAttention (Xu et al., 2025) uses the sum of anti-diagonal elements within each block as a low-cost proxy, FlexPrefill (Lai et al., 2025) exploits the attention pattern of the last query block to select important vertical and slash structures, and SpargeAttention (Zhang et al., 2025) adopts an online predict-then-verify scheme with similarity-gated block compression and softmax-aware kernel filtering. These approaches achieve high throughput, but their effectiveness hinges on alignment between the heuristic and data modality. In complex multimodal tasks, the assumption that a single block can represent global context often fails, leading to sub-optimal sparsity and accuracy degradation.

### 3.3. Discussion and Open Question

Existing inference-time dynamic approaches exhibit a persistent trade-off: high-fidelity proxy methods achieve high accuracy but lack practicality and generality due to com-

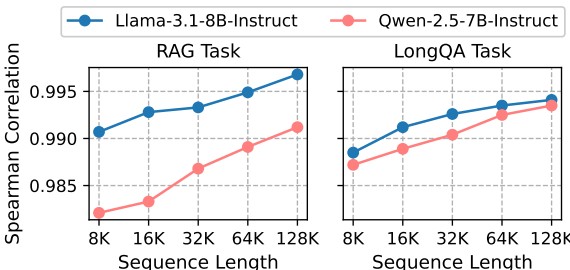

*Figure 1.* Spearman correlation between compressed and original block importance rankings on HELMET.

plexity and model-specific preprocessing, while low-cost heuristic approaches are efficient but fragile, relying on oversimplified, modality-specific assumptions. This tension motivates our guiding question: *Is there a proxy paradigm that is structurally simple and hardware-friendly, yet robust enough to generate effective sparse patterns across diverse modalities—without model training or adaptation?*

## 4. Design

This section introduces UniSparse, a unified mechanism for dynamic sparse attention. We begin with the core intuition (§4.1), followed by a description of the compression mechanism (§4.2) and the dynamic block selection algorithm (§4.3), and conclude with a complexity analysis (§4.4).

### 4.1. Intuition and Overview

**Key Insight: Composite Tokens as Universal Summaries.** The central hypothesis of UniSparse is that *semantically meaningful attention patterns can be reliably approximated in a drastically compressed token space*. Rather than computing importance scores at native token granularity or relying on partial evaluations that sample only a subset of query-key interactions, we propose to construct ***composite tokens***—coarse-grained summary vectors obtained through spatial pooling of fine-grained tokens. The intuition is rooted in the observation that tokens within local neighborhoods often exhibit semantic coherence, making their aggregate representation a faithful proxy for the collective importance of the underlying tokens.

Formally, consider a query block $\mathbf{Q}_i \in \mathbb{R}^{S \times d_k}$ and key block $\mathbf{K}_j \in \mathbb{R}^{S \times d_k}$. Instead of computing the full $S \times S$ attention score matrix, we compress them into composite representations $\tilde{\mathbf{Q}}_i \in \mathbb{R}^{S' \times d_k}$ and $\tilde{\mathbf{K}}_j \in \mathbb{R}^{S' \times d_k}$ where $S' \ll S$, compute a much smaller $S' \times S'$ score matrix, and use block-wise aggregated scores to determine block importance.

To validate this hypothesis, we measure the Spearman correlation between block importance rankings computed in compressed space (compression ratio $c = 8$) versus original token-level rankings on HELMET (Yen et al., 2025). Fig-

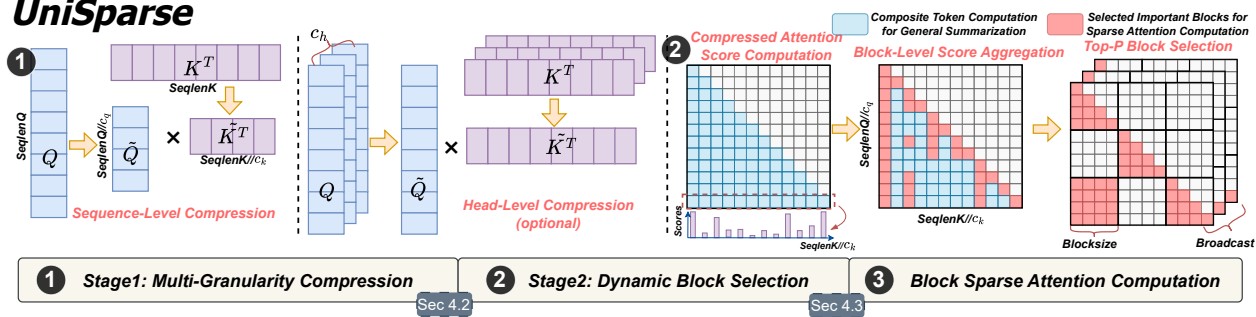

*Figure 2.* Workflow of UniSparse for efficient sparse attention. Given query, key, and value tensors, UniSparse proceeds in three stages: (❶) **Multi-Granularity Compression** aggregates fine-grained tokens into composite tokens along the sequence and (optionally) head dimensions; (❷) **Dynamic Block Selection** computes proxy attention scores in the compressed space, aggregates them to block-level importance, and identifies the salient query–key block pairs via Top-$P$ filtering to form the sparse mask $\mathcal{M}$; (❸) **Block-Sparse Attention Computation** then executes the final attention restricted by $\mathcal{M}$ using standard block-sparse FlashAttention kernels. We omit the internal details of stage ❸ in this figure, as it follows the standard block-sparse attention paradigm.

ure 1 shows consistently high correlation ($\rho > 0.98$) across different sequence lengths on both RAG and LongQA tasks for Llama and Qwen models, while reducing proxy computation by orders of magnitude. This strong rank preservation validates our core insight: block selection fundamentally requires accurate *relative ordering* rather than absolute scores, as we elaborate in the following analysis (§4.2).

Crucially, this compression-based proxy is *modality-agnostic*. Unlike methods that rely on domain-specific heuristics or structural assumptions about specific data patterns, spatial pooling is a universal operation applicable to any sequential data, be it text tokens, video frame patches, or audio segments. This universality is the foundation of UniSparse's cross-modality effectiveness.

**Overview.** As shown in Figure 2, UniSparse operates in two stages. In *multi-granularity compression* (❶), query and key matrices are compressed along the sequence dimension via average pooling, transforming fine-grained tokens into coarser composite tokens. In *dynamic block selection* (❷), attention scores are computed in compressed space, aggregated at block level, and filtered through Top-$P$ to identify important blocks. The resulting sparse mask $\mathcal{M}$ guides block-sparse attention (❸).

### 4.2. Multi-Granularity Compression

**Sequence-Level Compression.** Given a sequence length $L$ with $N = L/S$ blocks of size $S$, we define compression factors $c_q$ for queries and $c_k$ for keys. Compressed sequence lengths are $L'_q = L/c_q$, $L'_k = L/c_k$, yielding $N'_q = L'_q/S$, $N'_k = L'_k/S$ compressed blocks. For the $i$-th query block $\mathbf{Q}_i = [\mathbf{q}_{i,1}, \ldots, \mathbf{q}_{i,S}]^T \in \mathbb{R}^{S \times d_k}$, we construct compressed block $\tilde{\mathbf{Q}}_{i'} \in \mathbb{R}^{S/c_q \times d_k}$ through average pooling:

$$\tilde{\mathbf{q}}_{i',t'} = \frac{1}{c_q} \sum_{m=0}^{c_q-1} \mathbf{q}_{i,t' \cdot c_q + m}, \quad t' = 1, 2, \ldots, S/c_q \quad (2)$$

where $i' = \lfloor i/(c_q/S) \rfloor$ maps original to compressed block indices. The compressed query matrix is $\tilde{\mathbf{Q}} \in \mathbb{R}^{L'_q \times d_k}$; similarly, $\tilde{\mathbf{K}} \in \mathbb{R}^{L'_k \times d_k}$ using factor $c_k$.

**Head-Level Compression (Optional).** To further reduce overhead, we introduce optional head compression with factor $c_h$. For $H$ attention heads, we group every $c_h$ consecutive heads and compute their average:

$$\tilde{\mathbf{Q}}^{(h')} = \frac{1}{c_h} \sum_{m=0}^{c_h-1} \tilde{\mathbf{Q}}^{(h' \cdot c_h + m)}, \quad h' = 0, 1, \ldots, \frac{H}{c_h} - 1 \quad (3)$$

This yields $H' = H/c_h$ compressed heads, likewise for keys. The sparse pattern from compressed heads is *broadcast* to all original heads within each group, maintaining per-head expressiveness while amortizing selection cost. This distills consensus importance signals across heads, with effectiveness varying by task and model.

**Why Compression Preserves Importance.** The effectiveness of compression in block importance estimation stems from a key insight: block selection is essentially a *ranking* rather than a *scoring* problem. As long as compression preserves the relative ordering of block importance, selection quality remains high. Average pooling over composite tokens naturally maintains this ordering by summarizing local semantics into representative vectors that reflect the collective relevance of fine-grained tokens. Our experiments (§5) confirm that such compression-based rankings closely match those derived from full token-level computations. A formal analysis—establishing (i) exact pre-softmax rank preservation, (ii) a Taylor-expansion bound on post-softmax distortion, and (iii) the centroid-optimality of average pooling—is deferred to Appendix A.

### 4.3. Dynamic Block Selection

**Compressed Attention Score Computation.** Using compressed queries $\tilde{\mathbf{Q}} \in \mathbb{R}^{L'_q \times d_k}$ and keys $\tilde{\mathbf{K}} \in \mathbb{R}^{L'_k \times d_k}$, we compute the attention score matrix in the compressed

space $\tilde{\mathbf{P}} = \frac{\tilde{\mathbf{Q}}\tilde{\mathbf{K}}^T}{\sqrt{d_k}} \in \mathbb{R}^{L'_q \times L'_k}$. This matrix has size $(L/c_q) \times (L/c_k)$, which is a factor of $c_q \cdot c_k$ smaller than the full $L \times L$ score matrix. We apply row-wise $\texttt{softmax}$ to obtain $\tilde{\mathbf{A}} = \texttt{softmax}(\tilde{\mathbf{P}}) \in \mathbb{R}^{L'_q \times L'_k}$, which serves as our lightweight proxy for block importance estimation.

**Block-Level Score Aggregation.** To determine important blocks in the *original* space, we aggregate compressed attention scores at block granularity. For each query block $\mathbf{Q}_i$, we identify its corresponding region in the compressed space and sum attention scores to each key block $\mathbf{K}_j$:

$$\texttt{Score}(i,j) = \sum_{t' \in \mathcal{R}_q(i)} \sum_{s' \in \mathcal{R}_k(j)} \tilde{\mathbf{A}}_{t',s'} \quad (4)$$

where $\mathcal{R}_q(i)$ and $\mathcal{R}_k(j)$ denote compressed token indices corresponding to original query block $i$ and key block $j$:

$$\mathcal{R}_x(u) = \left\{ z' \mid \left\lfloor \frac{z' \cdot c_x}{S} \right\rfloor = u \right\}, \quad x \in \{q, k\} \quad (5)$$

where $u = i$ and $z' = t'$ when $x = q$, and $u = j$, $z' = s'$ when $x = k$. This produces a block-level score matrix $\textbf{Score} \in \mathbb{R}^{N \times N}$, where $N = L/S$ in the original space.

**Top-$P$ Block Selection.** For each query block $i$, we select key blocks by threshold $P \in (0, 1]$, representing the proportion of attention mass to retain. The mask is $\mathcal{M}_{ij} = 1$ if $j \in \mathcal{T}_P(i)$, else 0, where $\mathcal{T}_P(i)$ is constructed by sorting key blocks by $\texttt{Score}(i,j)$ in descending order and accumulating until:

$$\mathcal{T}_P(i) = \left\{ j \mid \frac{\sum_{j' \in \mathcal{T}_P(i)} \texttt{Score}(i,j')}{\sum_{j'=1}^{N} \texttt{Score}(i,j')} \geq P \right\} \quad (6)$$

The overall workflow is summarized in Algorithm 1. We discuss hardware-aware optimizations, including fused kernels and library integration, in Appendix B.

### 4.4. Complexity Analysis

Understanding the computational overhead of dynamic sparsity is critical for evaluating practical efficiency. The selection overhead $C_{\text{select}}$ of UniSparse decomposes as:

$$\underbrace{\mathcal{O}(Lhd_k)}_{\texttt{Compression}} + \underbrace{\mathcal{O}\left(\frac{L^2 hd_k}{c_q \cdot c_k \cdot c_h}\right)}_{\textbf{Compressed QK}} + \underbrace{\mathcal{O}\left(\frac{L^2 h}{c_q \cdot c_k \cdot c_h}\right)}_{\texttt{softmax\&Aggregation}} + \underbrace{\mathcal{O}(N^2 h \log N)}_{\texttt{Top-}P}$$
$$(7)$$

where $h$ is the number of attention heads. The compressed attention score computation dominates with complexity $\mathcal{O}(L^2 hd_k/(c_q \cdot c_k \cdot c_h))$. The $\texttt{Top-}P$ selection has complexity $\mathcal{O}((L^2 h/S^2) \log(L/S))$, which is negligible for reasonably large block sizes (e.g., $S = 128$).

Table 1 compares UniSparse with existing methods. UniSparse achieves **global** evaluation—considering all query-key interactions—through compression, unlike **local** methods (MInference, FlexPrefill) that only sample the last query

---

**Algorithm 1** UniSparse: Multi-Granularity Block Selection

**Input:** $\boldsymbol{Q}, \boldsymbol{K}, \boldsymbol{V} \in \mathbb{R}^{L \times d_k}$, block size $S$, compression factors $c_q, c_k, c_h$, threshold $P \in (0, 1]$

*# Step 1: Multi-Granularity Compression*
$\tilde{\boldsymbol{Q}}, \tilde{\boldsymbol{K}} \leftarrow \texttt{AvgPool}_{\text{seq}}(\boldsymbol{Q}, c_q), \texttt{AvgPool}_{\text{seq}}(\boldsymbol{K}, c_k)$
**if** $c_h > 1$ **then**
    $\tilde{\boldsymbol{Q}}, \tilde{\boldsymbol{K}} \leftarrow \texttt{AvgPool}_{\text{head}}(\tilde{\boldsymbol{Q}}, c_h), \texttt{AvgPool}_{\text{head}}(\tilde{\boldsymbol{K}}, c_h)$
**end if**

*# Step 2: Compressed Attention Computation*
$\tilde{\boldsymbol{P}} \leftarrow \tilde{\boldsymbol{Q}}\tilde{\boldsymbol{K}}^T/\sqrt{d_k}$
$\tilde{\boldsymbol{A}} \leftarrow \texttt{softmax}(\tilde{\boldsymbol{P}})$

*# Step 3: Block-Level Score Aggregation*
$\boldsymbol{Score} \leftarrow \texttt{BlockAggregate}(\tilde{\boldsymbol{A}}, S, c_q, c_k)$

*# Step 4: Top-P Block Selection with Causal Mask*
**for** $i = 1$ to $N$ **do**
    $\boldsymbol{Score}[i, j] \leftarrow -\infty$ for all $j > i$
    $\mathcal{T}_P(i) \leftarrow \texttt{TopP\_Select}(\boldsymbol{Score}[i, :], P)$
    $\mathcal{M}[i, \mathcal{T}_P(i)] \leftarrow 1$
**end for**
**if** $c_h > 1$ **then**
    $\mathcal{M} \leftarrow \texttt{Broadcast}_{\text{head}}(\mathcal{M}, c_h)$
**end if**

*# Step 5: Block-Sparse Attention Computation*
$\boldsymbol{O} \leftarrow \texttt{SparseFlashAttention}(\boldsymbol{Q}, \boldsymbol{K}, \boldsymbol{V}, \mathcal{M})$
**Output:** $\boldsymbol{O} \in \mathbb{R}^{L \times d_k}$

---

*Table 1.* Complexity and evaluation scope of dynamic sparse attention methods.

| Method | Selection Complexity $C_{\text{select}}$ | Evaluation Scope |
|---|---|---|
| MInference | $\mathcal{O}(S \cdot L \cdot h \cdot d_k) + \mathcal{O}_{\text{offline}}$ | *Local (Last Query Block)* |
| FlexPrefill | $\mathcal{O}(S \cdot L \cdot h \cdot d_k)$ | *Local (Last Query Block)* |
| XAttention | $\mathcal{O}(L^2 hd_k/\text{stride})$ | *Global (Strided Sampling)* |
| **UniSparse** | $\mathcal{O}(L^2 hd_k/(c_q c_k c_h))$ | *Global (Compressed Space)* |

block. Compared to XAttention's strided sampling, UniSparse achieves lower overhead via multi-granularity compression while maintaining better coverage of semantic dependencies. With appropriate $(c_q, c_k, c_h)$, UniSparse's overhead approaches local methods while preserving global evaluation benefits (§5.3).

## 5. Experiments

This section evaluates UniSparse across models and workloads and addresses three key questions: (i) Can UniSparse generalize across models and modalities? (ii) Does UniSparse achieve hardware efficiency? (iii) How does each design choice contribute to the overall results?

### 5.1. Experimental Setup

All experiments are run on a single NVIDIA H200 GPU. For fairness, all methods apply sparse attention computation only during the prefill stage, while maintaining full KV cache and dense computation during decode stage.

**Models.** We evaluate UniSparse on three representative

*Table 2.* Performance comparison on RULER across context lengths from 4K to 128K tokens. We report accuracy scores at each length and average sparsity ratios. The best results among sparse methods at similar sparsity levels are highlighted.

| Model | Method | Sparsity (↑) | 4K | 8K | 16K | 32K | 64K | 128K | Overall (↑) |
|---|---|---|---|---|---|---|---|---|---|
| *Llama-3.1-8B-Instruct* | FlashAttention | - | 97.30 | 96.95 | 96.42 | 93.96 | 89.96 | 82.65 | 92.87 |
| | FlexPrefill (0.95) | 64.16% | 95.58 | 96.29 | 95.79 | **94.38** | **90.89** | **81.31** | 92.37 |
| | XAttention (0.9) | 56.44% | 97.21 | 96.57 | 95.85 | 93.73 | 89.02 | 79.17 | 91.93 |
| | **UniSparse (0.9)** | 58.90% | **97.38** | **96.59** | **96.43** | 94.25 | 89.29 | 80.73 | **92.44** |
| | MInference | 50.39% | 97.29 | 96.69 | 96.71 | 93.40 | 88.67 | 81.38 | 92.35 |
| | SpargeAttention | 47.49% | **97.45** | 95.60 | 94.80 | 93.21 | 87.35 | 79.23 | 91.27 |
| | FlexPrefill (0.99) | 36.51% | 97.33 | **96.79** | **96.35** | 93.99 | **90.53** | 80.52 | **92.59** |
| | XAttention (0.95) | 44.33% | 97.27 | 96.49 | 96.15 | 93.82 | 90.16 | 81.34 | 92.54 |
| | **UniSparse (0.95)** | 46.33% | 97.16 | 96.52 | 96.31 | **94.03** | 89.40 | **81.94** | 92.56 |
| *Qwen2.5-7B-Instruct* | FlashAttention | - | 94.77 | 89.42 | 87.84 | 85.46 | 81.18 | 70.92 | 84.93 |
| | FlexPrefill (0.95) | 64.14% | 83.78 | 81.45 | 80.20 | 76.50 | 71.21 | 58.06 | 75.20 |
| | XAttention (0.9) | 58.17% | 92.84 | 87.91 | 85.02 | 82.80 | 77.14 | 66.58 | 82.05 |
| | **UniSparse (0.9)** | 60.12% | **93.23** | **87.69** | **85.99** | **84.26** | **79.25** | **68.00** | **83.07** |
| | MInference | 39.32% | **94.68** | **89.65** | **88.56** | 85.04 | 78.47 | 65.08 | 83.58 |
| | SpargeAttention | 47.51% | 87.15 | 85.62 | 84.94 | 83.43 | 78.67 | 67.42 | 81.21 |
| | FlexPrefill (0.99) | 37.75% | 92.42 | 88.77 | 85.91 | 84.13 | 77.89 | 67.69 | 82.80 |
| | XAttention (0.95) | 46.44% | 93.87 | 88.36 | 86.84 | 84.15 | 79.04 | **69.51** | 83.63 |
| | **UniSparse (0.95)** | 47.25% | 94.31 | 88.96 | 87.19 | **85.23** | **80.42** | 69.47 | **84.26** |

*Table 3.* Performance comparison on HELMET across context lengths from 8K to 128K tokens (metrics follows Table 2).

| Model | Method | Sparsity (↑) | 8K | 16K | 32K | 64K | 128K | Overall(↑) |
|---|---|---|---|---|---|---|---|---|
| *Llama-3.1-8B-Instruct* | FlashAttention | - | 61.06 | 58.73 | 56.57 | 55.92 | 49.79 | 56.41 |
| | FlexPrefill (0.95) | 65.92% | 55.89 | 54.73 | 53.14 | 50.45 | 48.42 | 52.53 |
| | XAttention (0.9) | 59.80% | 59.81 | 57.40 | 55.55 | 53.74 | 47.74 | 54.85 |
| | **UniSparse (0.9)** | 61.82% | **60.86** | **57.70** | **56.21** | **54.24** | **48.91** | **55.58** |
| | MInference | 56.34% | **61.44** | 57.96 | 55.49 | 53.69 | 48.29 | 55.37 |
| | SpargeAttention | 48.10% | 58.15 | 57.09 | **56.43** | 54.43 | 49.19 | 55.06 |
| | FlexPrefill (0.99) | 38.52% | 60.66 | 58.05 | 56.22 | **55.44** | 49.68 | 56.01 |
| | XAttention (0.95) | 47.45% | 60.76 | 57.80 | 55.92 | 54.75 | 49.37 | 55.72 |
| | **UniSparse (0.95)** | 48.65% | 61.37 | **58.34** | 56.22 | 55.27 | **50.22** | **56.21** |
| *Qwen2.5-7B-Instruct* | FlashAttention | - | 55.10 | 50.54 | 48.51 | 44.85 | 39.33 | 47.67 |
| | FlexPrefill (0.95) | 63.03% | 44.23 | 40.79 | 39.87 | 36.28 | 30.06 | 38.25 |
| | XAttention (0.9) | 59.08% | 52.46 | 47.99 | 45.63 | 43.17 | 36.92 | 45.23 |
| | **UniSparse (0.9)** | 60.84% | **53.13** | **48.62** | **46.93** | **44.22** | **38.11** | **46.20** |
| | MInference | 46.95% | **54.91** | **50.36** | 46.93 | 42.62 | 34.05 | 45.77 |
| | SpargeAttention | 48.15% | 50.34 | 48.88 | 47.19 | 44.04 | 38.34 | 45.76 |
| | FlexPrefill (0.99) | 37.75% | 52.77 | 48.91 | 46.39 | 43.06 | 36.60 | 45.55 |
| | XAttention (0.95) | 47.31% | 53.93 | 49.59 | 47.10 | 44.65 | 38.19 | 46.69 |
| | **UniSparse (0.95)** | 47.51% | 53.91 | 49.96 | **47.76** | **44.95** | **38.58** | **47.03** |

*Table 4.* Performance comparison on Video-MME using Qwen2.5-VL-7B-Instruct. We report accuracy on short (< 2 min), medium (4-15 min), and long (> 30 min) videos, with and without subtitle inputs.

| Method | Without Subtitles | | | | | With Subtitles | | | | |
|---|---|---|---|---|---|---|---|---|---|---|
| | Sparsity(↑) | Short | Medium | Long | Overall(↑) | Sparsity(↑) | Short | Medium | Long | Overall(↑) |
| FlashAttention | - | 75.1 | 66.7 | 54.2 | 65.3 | - | 75.3 | 71.7 | 61.7 | 69.6 |
| FlexPrefill (0.95) | 57.9% | 73.9 | 65.8 | **53.7** | 64.4 | 66.3% | **75.2** | 69.9 | 61.7 | 68.9 |
| XAttention (0.9) | 60.2% | 73.6 | 63.6 | 51.9 | 63.0 | 61.1% | 74.0 | 69.8 | 60.4 | 68.1 |
| **UniSparse (0.9)** | 57.7% | **74.4** | **65.9** | 53.3 | **64.6** | 58.6% | 74.9 | **71.2** | **63.4** | **69.9** |
| FlexPrefill (0.99) | 27.3% | 74.6 | **66.8** | 53.7 | **65.0** | 36.3% | **75.4** | **71.7** | 61.7 | 69.6 |
| XAttention (0.95) | 49.2% | 74.1 | 64.7 | 52.8 | 63.9 | 50.4% | 75.2 | 70.8 | 61.1 | 69.0 |
| **UniSparse (0.95)** | 45.9% | **74.8** | 65.8 | **54.3** | 65.0 | 47.2% | 75.3 | **71.7** | **62.0** | **69.7** |

models to assess generality across architectures. *(1) Meta-Llama-3.1-8B-Instruct* (Meta-Llama-3.1-8B-Instruct): a decoder-only model supporting 128K tokens via RoPE scaling. *(2) Qwen2.5-7B-Instruct* (Qwen2.5-7B-Instruct): extends context to 128K using YaRN interpolation with strong multilingual capability. *(3) Qwen2.5-VL-7B-Instruct* (Qwen2.5-VL-7B-Instruct): a unified vision-language model handling both text and video inputs. These models cover diverse positional encodings and modalities.

**Workloads.** We employ three challenging long-context benchmarks spanning synthetic to real-world tasks and text to video understanding. (1) RULER (Hsieh et al., 2024): a synthetic benchmark with controllable complexity, testing retrieval, aggregation, and multi-hop reasoning over sequences up to 128K tokens. (2) HELMET (Yen et al., 2025): a real-world benchmark spanning seven application categories (e.g., long-document QA, summarization, RAG), requiring deep semantic understanding of documents up to 128K tokens. (3) Video-MME (Fu et al., 2025): evaluates video understanding over 6 visual domains and 30 subfields with multi-modal inputs (frames, subtitles, audio), testing temporal and cross-modal reasoning.

**Baselines and Configuration.** UniSparse applies multi-granularity compression with global evaluation, using $c_q=c_k=8$ and Top-$P$ thresholds $P=0.9, 0.95$ by default; additional settings are analyzed in Section 5.4. We compare against five baselines under comparable sparsity levels, following official parameter recommendations denoted as $\tau$ and $\gamma$. (1) FlashAttention (Dao et al., 2022): full attention baseline providing the accuracy upper bound. (2) MInference (Jiang et al., 2024): offline pattern search with head-specific sparse configurations; we use official searched files for each model. (3) FlexPrefill (Lai et al., 2025): lightweight local evaluation using the last Q-block as probe, with $\tau=0.1$ and $\gamma=0.95, 0.99$. (4) XAttention (Xu et al., 2025): global evaluation via anti-diagonal scoring with strided sampling (stride=8), with $\tau=0.9, 0.95$. (5) SpargeAttention (Zhang et al., 2025): training-free online predict-then-verify scheme with similarity-gated compression and softmax-aware kernel filtering, using topk=0.3. Comparisons are organized into **moderate sparsity** (FlexPrefill-0.99, XAttention-0.95, SpargeAttention) and **high sparsity** (FlexPrefill-0.95, XAttention-0.9).

### 5.2. Main Results

Tables 2–4 summarize results on text benchmarks (RULER, HELMET) using Llama and Qwen, and on video benchmark (Video-MME) using Qwen-VL. Across all modalities and context lengths (up to 128K tokens or long-duration videos), UniSparse consistently achieves superior accuracy–efficiency trade-offs.

**Synthetic Long-Context Tasks.** On RULER, UniSparse achieves the best overall performance among sparse methods across both sparsity setting. On Llama, UniSparse-0.95 achieves 92.56 accuracy, matching FlexPrefill-0.99 at 92.59 while maintaining higher sparsity (46.33% vs. 36.51%) and surpassing SpargeAttention (91.27). On Qwen, UniSparse-0.95 significantly outperforms all baselines with 84.26 accuracy compared to MInference's 83.58, XAttention-0.95's 83.63, and SpargeAttention's 81.21. UniSparse demonstrates consistent performance across all context lengths

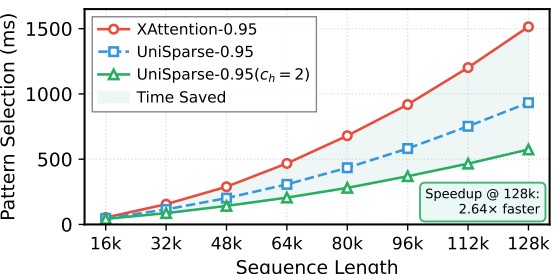

*Figure 3.* Block selection speedup across sequence lengths.

from 4K to 128K tokens, with particularly strong results at extreme lengths where long-range dependency modeling is critical.

**Real-World Semantic Understanding.** UniSparse's advantages become more pronounced on HELMET's real-world tasks requiring complex semantic understanding. On Llama, UniSparse-0.95 achieves 56.21 accuracy with 48.65% sparsity, outperforming FlexPrefill-0.99 (56.01), MInference (55.37), and SpargeAttention (55.06). On Qwen, UniSparse-0.95 achieves 47.03 accuracy, surpassing MInference (45.77), XAttention-0.95 (46.69), and SpargeAttention (45.76). Consistent improvements across both models validate UniSparse's effectiveness in text-based long-context scenarios. Detailed task-level analysis is provided in Appendix C.

**Multi-Modal Video Understanding.** UniSparse demonstrates strong cross-modal reasoning in both subtitle configurations. Under high sparsity, UniSparse-0.9 achieves 64.6 accuracy (57.7% sparsity) without subtitles, outperforming FlexPrefill-0.95 (64.4) and significantly surpassing XAttention-0.9 (63.0). With subtitles, UniSparse-0.9 achieves 69.9 accuracy with 58.6% sparsity, even surpassing FlashAttention's full attention, demonstrating its ability to preserve critical cross-modal information under aggressive compression. These results highlight UniSparse's capability to handle temporal reasoning where important information is distributed across visual and textual modalities.

**Additional Real-World Reasoning Evaluation.** We further evaluate on LongBench v2 (Bai et al., 2025) (Llama-3.1-8B-Instruct), where UniSparse-0.95 achieves the highest overall accuracy (30.4) among all methods, even surpassing full-attention FlashAttention (29.8). Detailed results, including stratified scores by difficulty and context length, are provided in Appendix D.

**Overall Performance Summary.** Across all benchmarks spanning text and video modalities, UniSparse demonstrates exceptional accuracy retention while computing less than half of attention blocks, consistently retaining over 99% of FlashAttention's accuracy and even surpassing full attention in certain cases. Compared to existing sparse methods, UniSparse's multi-granularity compression enables more effective adaptive selection that captures both coarse and fine-

*Table 5.* Ablation on compression strategies: average sparsity and overall scores over 4K–128K context lengths for Llama and Qwen.

| Method | Meta-Llama-3.1-8B-Instruct | | | | Qwen2.5-7B-Instruct | | | |
|---|---|---|---|---|---|---|---|---|
| | Sparsity (↑) | RULER (↑) | Sparsity (↑) | HELMET (↑) | Sparsity (↑) | RULER (↑) | Sparsity (↑) | HELMET (↑) |
| **UniSparse** | 46.33% | **92.56** | 48.65% | **56.21** | 47.25% | **84.26** | 47.51% | **47.03** |
| Max | 53.93% | 92.11 | 56.85% | 54.83 | 56.09% | 81.01 | 58.98% | 43.46 |
| Stochastic | 51.68% | 92.21 | 54.44% | 55.23 | 54.21% | 80.84 | 56.39% | 44.05 |

*Table 6.* Ablation on Q-K compression ratio with fixed total compression ($c_q \times c_k = 64$) and varying allocation between queries and keys.

| Method | Meta-Llama-3.1-8B-Instruct | | | | Qwen2.5-7B-Instruct | | | |
|---|---|---|---|---|---|---|---|---|
| | Sparsity (↑) | RULER (↑) | Sparsity (↑) | HELMET (↑) | Sparsity (↑) | RULER (↑) | Sparsity (↑) | HELMET (↑) |
| $c_q = 32, c_k = 2$ | 48.32% | 91.37 | 49.75% | 54.88 | 49.90% | 81.93 | 49.94% | 45.87 |
| $c_q = 16, c_k = 4$ | 46.84% | 92.46 | 49.02% | 55.63 | 48.93% | 83.61 | 49.07% | 46.89 |
| **UniSparse** | 46.33% | 92.56 | 48.65% | **56.21** | 47.25% | **84.26** | 47.51% | 47.03 |
| $c_q = 4, c_k = 16$ | 46.32% | 92.54 | 48.67% | 55.99 | 47.48% | 84.11 | 47.81% | **47.27** |
| $c_q = 2, c_k = 32$ | 44.88% | **92.57** | 47.15% | 55.97 | 45.68% | 83.93 | 46.11% | 47.24 |

*Table 7.* Ablation on compression granularity with varying compression factor $c$ under balanced Q-K compression ($c_q = c_k = c$).

| Method | Meta-Llama-3.1-8B-Instruct | | | | Qwen2.5-7B-Instruct | | | |
|---|---|---|---|---|---|---|---|---|
| | Sparsity (↑) | RULER (↑) | Sparsity (↑) | HELMET (↑) | Sparsity (↑) | RULER (↑) | Sparsity (↑) | HELMET (↑) |
| $c = 4$ | 47.42% | 92.59 | 49.79% | 56.02 | 48.82% | **84.51** | 49.09% | 47.01 |
| **UniSparse** | 46.33% | 92.56 | 48.65% | **56.21** | 47.25% | 84.26 | 47.51% | **47.03** |
| $c = 16$ | 46.86% | **92.64** | 49.36% | 55.94 | 48.84% | 83.43 | 49.25% | 46.54 |
| $c = 32$ | 46.16% | 92.54 | 48.64% | 55.87 | 48.30% | 83.38 | 48.87% | 46.49 |

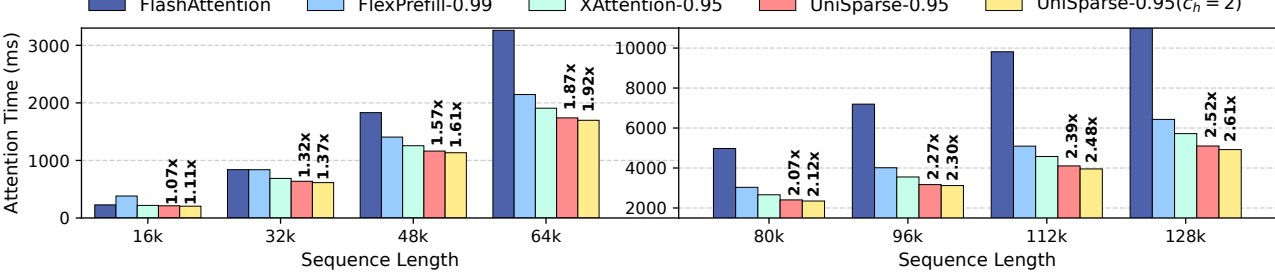

*Figure 4.* End-to-end attention speedup across sequence lengths (16K-128K tokens).

grained patterns, resulting in consistent advantages across synthetic reasoning, real-world semantic understanding, and multi-modal tasks without task-specific tuning.

### 5.3. Efficiency Results

We measure efficiency using Llama on HELMET, whose real-world tasks better reflect practical deployment.

**Block Selection Overhead.** Figure 3 compares selection overhead $C_{\text{select}}$ of global evaluation methods from 16K to 128K tokens. We exclude FlexPrefill and MInference due to their limited local evaluation scope (Table 1). UniSparse-0.95 achieves 1.62× speedup over XAttention-0.95 at 128K tokens. Unlike XAttention's fixed-stride sampling, UniSparse performs global evaluation incorporating all query and key tokens while reducing scoring complexity to $\mathcal{O}(L^2 h d_k/(c_q c_k c_h))$ through multi-granularity compression. With optional head compression ($c_h = 2$), UniSparse-0.95 achieves 2.64× speedup, further reducing redundant cross-head evaluations.

**End-to-End Attention Speedup.** Figure 4 presents end-to-end attention time including block selection and sparse com-

putation. At 128K tokens, UniSparse-0.95 achieves 2.52× speedup over FlashAttention while retaining over 99% accuracy. With head compression ($c_h = 2$), UniSparse reaches 2.61× speedup. We exclude MInference due to its expensive offline pattern search. UniSparse consistently outperforms FlexPrefill and XAttention across all sequence lengths. FlexPrefill-0.99, despite lower selection overhead from local evaluation, is slower overall due to imprecise block selection requiring higher sparsity. XAttention-0.95 achieves competitive attention time but incurs higher selection overhead, leading to longer total latency. These results demonstrate that UniSparse's balanced design—efficient multi-granularity compression combined with effective sparse pattern identification—delivers superior end-to-end efficiency while maintaining accuracy. Additional efficiency measurements on a different GPU architecture (NVIDIA H20), with sequence lengths extended to 256K and end-to-end time-to-first-token (TTFT) speedups, are reported in Appendix E.

### 5.4. Ablation Study

We conduct ablation studies to analyze key design choices in UniSparse, including compression strategies, Q-K alloca-

tion ratios, and compression granularity.

**Compression Strategy.** Table 5 compares different compression strategies for constructing composite tokens. UniSparse adopts average pooling (Eq. 2) as default. While max pooling and stochastic pooling achieve higher sparsity, they sacrifice accuracy due to incomplete information aggregation: max pooling retains only the most salient feature per window, while stochastic pooling selects a single token probabilistically, both failing to capture full contextual information. In contrast, average pooling preserves comprehensive information by uniformly aggregating all tokens, enabling more accurate block importance estimation.

**Q-K Compression Ratio Allocation.** Table 6 investigates different Q-K compression allocations while fixing the total factor ($c_q \times c_k = 64$). Balanced compression ($c_q = c_k = 8$) achieves the best overall performance. Query-biased compression (e.g., $c_q = 32, c_k = 2$) degrades significantly, as over-compressing queries loses fine-grained distinctions for accurate scoring. Key-biased compression (e.g., $c_q = 4, c_k = 16$) performs competitively in certain cases, particularly on Qwen, suggesting model-specific characteristics that warrant further investigation. Beyond accuracy, balanced compression enables more efficient implementations under causal masking, as symmetric Q-K granularity simplifies computation of valid attention ranges.

**Compression Granularity.** Table 7 examines compression factor $c$ with balanced Q-K compression ($c_q = c_k = c$). Smaller factors (e.g., $c = 4$) capture more nuanced patterns but incur higher selection overhead. Larger factors (e.g., $c = 16$) reduce cost but risk over-smoothing local variations. Our default $c = 8$ balances efficient global evaluation with sufficient granularity to distinguish important blocks, resulting in robust performance. UniSparse supports optional head-level compression ($c_h > 1$) to further reduce selection overhead; we analyze this trade-off in Appendix F.

## 6. Discussion and Limitations

UniSparse is designed for long-context inference, where the quadratic cost of full attention dominates. At shorter sequence lengths (e.g., $L \lesssim 16\text{K}$), the constant overhead of compression, proxy attention, and Top-$P$ selection is amortized less effectively, and the benefit of dynamic sparsification narrows relative to dense attention. In such regimes, a practical deployment can simply fall back to dense attention, recovering the long-context speedup whenever it is most needed. Our analysis in Section 4 also provides a leading-order characterization of how compression preserves block-importance ranking; deriving tight distribution-free worst-case bounds and extending UniSparse to other compression operators are promising directions for future work.

## 7. Conclusion

In this work, we introduce UniSparse, which leverages composite tokens as a universal summary for efficient sparse attention. Extensive evaluations across tasks and modalities show that UniSparse achieves state-of-the-art accuracy while reducing computation costs. By bridging efficiency and generality, UniSparse enables scalable long-context modeling and provides a practical foundation for future research in dynamic sparse attention.

## Impact Statement

This paper presents work whose goal is to advance the field of Machine Learning. There are many potential societal consequences of our work, none which we feel must be specifically highlighted here.

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

# A. Theoretical Analysis of Rank Preservation under Average Pooling

We provide a formal analysis of why average pooling preserves block importance ranking, addressing the question of when this approximation succeeds and when it may fail. The analysis is organized into three complementary results: (i) exact rank preservation in pre-softmax space, (ii) a moment-expansion bound on post-softmax distortion, and (iii) the centroid-optimality of average pooling.

## A.1. Setup and Notation

Let $\mathbf{Q}, \mathbf{K} \in \mathbb{R}^{L \times d_k}$ denote the original query and key matrices with sequence length $L$ and head dimension $d_k$. Each sequence is partitioned into non-overlapping blocks of size $S$ at the token level (used for sparse attention) and further compressed into composite tokens by averaging consecutive groups of $c_q$ (resp. $c_k$) tokens:

$$\tilde{q}_{t'} = \frac{1}{c_q} \sum_{m=0}^{c_q-1} q_{t'c_q+m}, \qquad \tilde{k}_{s'} = \frac{1}{c_k} \sum_{n=0}^{c_k-1} k_{s'c_k+n}.$$

For a pair of token-level blocks $(i, j)$ of size $S$, denote by $\mathrm{Score}(i, j)$ the sum of pre-softmax logits over all token pairs in the block, and by $\widehat{\mathrm{Score}}(i, j)$ the same quantity computed on composite tokens.

## A.2. Result 1: Pre-Softmax Exact Rank Preservation

**Proposition** (Pre-softmax). *For any pair of blocks $(i, j)$,*

$$\widehat{\mathrm{Score}}(i, j) = \frac{1}{c_q c_k} \mathrm{Score}(i, j),$$

*and consequently the ranking induced by $\widehat{\mathrm{Score}}$ is identical to that induced by $\mathrm{Score}$.*

*Proof.* Bilinearity of the inner product yields

$$\tilde{q}_{t'}^{\top} \tilde{k}_{s'} = \frac{1}{c_q c_k} \sum_{m=0}^{c_q-1} \sum_{n=0}^{c_k-1} q_{t'c_q+m}^{\top} k_{s'c_k+n}.$$

Summing over all compressed pairs within block $(i, j)$ recovers $\frac{1}{c_q c_k}$ times the full sum, since average pooling partitions tokens without overlap. The scaling factor $\frac{1}{c_q c_k}$ is independent of $(i, j)$, so rankings are exactly preserved. $\square$

In other words, in the absence of softmax, average pooling incurs zero information loss for block ranking.

## A.3. Result 2: Post-Softmax Moment-Expansion Bound

Softmax introduces nonlinearity through the exponential. Let $Z = q_m^{\top} k_n / \sqrt{d_k}$ denote a scaled dot product over a

uniformly random token pair within block $(i, j)$. The true block importance is proportional to $\mathbb{E}[\exp Z]$, whereas our proxy computes $\exp(\mathbb{E}[Z])$.

By Taylor expansion of $\exp(\cdot)$ around $\mu = \mathbb{E}[Z]$,

$$\mathbb{E}[\exp Z] = \exp(\mathbb{E}[Z]) \cdot \left(1 + \tfrac{1}{2}\mathrm{Var}(Z) + \tfrac{1}{6}\mathbb{E}[(Z - \mu)^3] + \cdots\right).$$

Higher-order terms are $\mathcal{O}(\sigma^3)$ and negligible when $\mathrm{Var}(Z)$ is small; the dominant correction is $\mathrm{Var}(Z)/2$.

**Rank Inversion Condition.** For two blocks $j_1$ (semantically more relevant) and $j_2$ with mean-logit gap $\Delta = \mathbb{E}[Z_{i,j_1}] - \mathbb{E}[Z_{i,j_2}] > 0$, the proxy ranking matches the true ranking unless

$$\exp(\Delta) < \frac{1 + \tfrac{1}{2}\mathrm{Var}(Z_{i,j_2})}{1 + \tfrac{1}{2}\mathrm{Var}(Z_{i,j_1})}. \tag{8}$$

**Quantitative Variance Bound.** Decompose $q_m = \bar{q} + \delta_m^q$, $k_n = \bar{k} + \delta_n^k$. Since intra-block deviations have zero mean and the pair is sampled independently, cross terms vanish and

$$\mathrm{Var}(Z) = \frac{\bar{k}^{\top}\Sigma_q \bar{k} + \bar{q}^{\top}\Sigma_k \bar{q}}{d_k} + \mathcal{O}\left(\frac{\sigma_q^2 \sigma_k^2}{d_k}\right),$$

where $\Sigma_q = \frac{1}{c_q} \sum_m \delta_m^q (\delta_m^q)^{\top}$ with $\mathrm{tr}(\Sigma_q) = \sigma_q^2$, and similarly for $\Sigma_k$. With LayerNorm encouraging approximately isotropic deviations, $\lambda_{\max}(\Sigma_q) \approx \sigma_q^2/d_k$, hence

$$\mathrm{Var}(Z) \leq \frac{\sigma_q^2 + \sigma_k^2}{d_k} \leq 2(1 - \cos\theta),$$

where $\cos\theta$ is the average pairwise cosine similarity of token representations within the block; the second inequality follows from LayerNorm, since $\sigma^2 = d_k - \|\bar{q}\|^2 \leq d_k(1 - \cos\theta)$.

**Numerical Interpretation.** For modestly coherent windows ($\cos\theta = 0.9$), $\mathrm{Var}(Z) \leq 0.2$ and the inversion condition in Eq. (8) reduces to $\Delta < \ln(1.1) \approx 0.095$. That is, only blocks whose true importance differs by less than $0.095$ in mean logit can swap ranks. In practice, important and unimportant blocks are separated by margins far exceeding this threshold, as evidenced by the $\geq 99\%$ accuracy retention at Top-$P = 0.95$ reported in Tables 2–4. Independent results on softmax-attention sensitivity (Kim et al., 2021) corroborate this: the Lipschitz constant of softmax attention scales with the weighted input variance, so bounded intra-block variance directly constrains the local rate of change. Our Spearman correlation analysis ($\rho > 0.98$, Figure 1) provides end-to-end empirical confirmation.

### A.4. Result 3: Centroid Optimality of Average Pooling

**Proposition** (Centroid optimality). *Among all single-vector summaries $u \in \mathbb{R}^{d_k}$ of a block $\{k_1, \ldots, k_S\}$, the mean $\bar{k} = \frac{1}{S} \sum_{n=1}^{S} k_n$ minimizes the squared-error objective $\sum_{n=1}^{S} \|k_n - u\|^2$.*

This is the standard MSE-optimality of the centroid in vector quantization. Because the dot product is linear in each operand, MSE-optimal input preservation translates into optimal preservation of the dot-product-based attention score. This explains why average pooling outperforms max pooling—which retains only one extreme token and discards the remaining $S - 1$—and stochastic pooling—which suffers from sampling variance—as observed in Table 5.

### A.5. Summary

Our analysis establishes that average pooling (i) preserves block ranking *exactly* in pre-softmax space, (ii) introduces a *well-bounded* post-softmax distortion whose rank inversions are restricted to near-equal-importance blocks—a regime in which selection outcomes are largely interchangeable—and (iii) is *MSE-optimal* as a centroid compressor. We acknowledge that a small fraction of near-tied blocks may occasionally swap ranks; empirically, $\rho > 0.98$ and $\geq 99\%$ accuracy at Top-$P = 0.95$ confirm that this has negligible impact. Tighter distribution-free worst-case bounds remain an interesting direction for future theoretical work.

## B. Hardware-Aware Implementation

**Library and Integration.** UniSparse is a standalone sparse attention library exposing a simple `unisparse_attn` API, that accepts $(\mathbf{Q}, \mathbf{K}, \mathbf{V})$ and returns $\mathbf{O}$. All complexity—including token selection, compression, and block-sparse computation—is encapsulated, enabling a drop-in replacement of standard attention layers. UniSparse supports arbitrary compression factors $(c_q, c_k, c_h)$ with guaranteed causal correctness for auto-regressive models, and employs kernel templating and auto-tuning to enable aggressive thread-level parallelization at higher compression ratios, thereby optimizing GPU occupancy across configurations. This enables seamless integration with existing frameworks (e.g., PyTorch, Transformers (Wolf et al., 2020)) without requiring model architecture changes or retraining. Combined with orthogonal techniques—KV cache compression and quantization (Li et al., 2024; Tang et al., 2024; Li et al., 2025), decoding-stage adaptive token pruning (Lin et al., 2025), and post-hoc sparse-output correction (Willette et al., 2025)—UniSparse enables efficient end-to-end long-context inference.

**Fused Kernel Optimization.** Algorithm 1 involves multiple sequential operations (e.g., `compression`, `softmax`,

`Top-P`) that would incur substantial memory traffic if implemented naively. Our fused kernel design keeps intermediate results in on-chip memory (shared memory and registers) rather than global HBM, combining these operations into unified kernel launches to minimize data movement. For causal masking, we precompute valid attention ranges and integrate masking directly into scoring kernels. These optimizations ensure $C_{\text{select}}$ remains a small fraction of end-to-end latency. For sparse attention computation, we integrate with block-sparse FlashAttention kernels that skip pruned blocks while maintaining memory efficiency.

## C. Detailed Analysis on HELMET Subtasks

Tables 8 and 9 present detailed performance across HELMET's six subtasks, evaluating diverse aspects of long-context understanding from 8K to 128K tokens. We analyze each task category below.

**Retrieval-Augmented Generation (RAG).** This task evaluates the ability to identify and utilize relevant passages from retrieved documents. UniSparse-0.95 maintains competitive performance with FlashAttention across all sequence lengths on both models—achieving 59.21 vs. 58.67 at 128K on Llama and 46.88 vs. 46.75 at 128K on Qwen. The compressed global evaluation effectively captures document-level relevance patterns, consistently outperforming other sparse methods at comparable sparsity levels.

**Passage Re-Ranking (ReRank).** Re-ranking requires fine-grained discrimination between candidate passages. UniSparse-0.95 demonstrates strong performance at moderate lengths, particularly excelling at 16K-32K tokens where it achieves 52.32 and 40.80 on Llama versus XAttention-0.95's 50.21 and 40.04, and FlexPrefill-0.99's 50.31 and 43.53. On Qwen, UniSparse-0.95 leads consistently at longer sequences (26.26 at 32K, 14.25 at 64K, 8.34 at 128K), validating that Top-$P$ selection adapts effectively to fine-grained relevance signals.

**Generation with Citations (Cite).** Citation tasks demand precise localization of information sources. UniSparse-0.95 maintains competitive accuracy across sequence lengths, leading at 32K-64K on Qwen with scores of 13.59 and 12.45 respectively. While all sparse methods face challenges at extreme lengths due to the inherent difficulty of exact positional tracking under compression, UniSparse's global evaluation preserves sufficient positional information for accurate source attribution at practical sparsity levels.

**Long-Document QA (LongQA).** Long-form question answering requires comprehensive reasoning across extended contexts. UniSparse-0.95 excels on this task, consistently achieving top performance among sparse methods—reaching 46.54 versus XAttention-0.95's 45.60 at 128K on Llama, and 43.19 versus XAttention-0.95's 42.17

at 128K on Qwen. The advantage becomes more pronounced as context length increases, demonstrating that global evaluation scope enables effective capture of long-range dependencies essential for complex reasoning.

**Many-Shot In-Context Learning (ICL).** ICL evaluates the ability to leverage examples within context for novel tasks. UniSparse-0.95 maintains consistently high and stable performance across all sequence lengths on both models, progressing from 71.16 to 84.36 across 8K to 128K on Llama, and from 69.52 to 79.88 on Qwen, closely matching FlashAttention's trajectory. This demonstrates that average pooling preserves semantic relationships between in-context examples and queries, enabling effective pattern recognition under high sparsity without degradation at extreme lengths.

**Synthetic Recall (Recall).** Recall tests precise information retrieval from long contexts. UniSparse shows strong performance particularly where other methods degrade significantly. UniSparse-0.9 achieves perfect 100.00 at 8K on Llama, and maintains robust recall at extreme lengths with 95.25 at 128K on Llama and 47.81 at 128K on Qwen, substantially outperforming XAttention-0.95's 91.56 and 45.56 respectively. The comprehensive global evaluation ensures critical information is not missed, unlike strided sampling that creates blind spots.

**Summary.** Across all six task categories and sequence lengths, UniSparse consistently demonstrates superior or competitive performance compared to existing sparse attention methods while maintaining higher sparsity levels. The multi-granularity compression enables efficient global evaluation that adapts to diverse attention patterns—from fine-grained re-ranking to long-range reasoning—without task-specific tuning. Notably, UniSparse's advantage becomes more pronounced at extreme sequence lengths, where it maintains stable performance while other methods experience significant degradation. Combined with its superior computational efficiency (§5.3)—achieving up to $2.61\times$ speedup over FlashAttention and consistently outperforming other sparse methods in both selection overhead and end-to-end latency—this consistent effectiveness validates our core insight: compression-based ranking preserves the semantic fidelity necessary for accurate block selection across multiple modalities and heterogeneous real-world tasks, establishing UniSparse as a unified and robust solution for long-context understanding.

## D. Additional Evaluation on LongBench v2

LongBench v2 (Bai et al., 2025) is a challenging long-context benchmark consisting of 503 multitask problems, stratified by difficulty (Easy/Hard) and context-length band (Short/Medium/Long). It is designed to assess deep understanding and reasoning over long contexts and complements

the synthetic retrieval focus of RULER and the application-oriented coverage of HELMET. We evaluate on Llama-3.1-8B-Instruct as a representative configuration.

As shown in Table 10, UniSparse-0.95 achieves the highest overall accuracy (30.4) among all methods, including the full-attention FlashAttention baseline (29.8) and the strongest sparse baselines (MInference 30.0, XAttention-0.95 29.0, FlexPrefill-0.97 27.4), while maintaining 66.7% sparsity. The advantage is most pronounced on the *Long* subset, precisely the regime where global compressed-space evaluation pays off. Notably, surpassing full attention is not a generic property of sparse methods: XAttention-0.95 and FlexPrefill-0.97 fall below full attention, while only MInference and UniSparse-0.95 outperform it. The contrast reflects how each baseline estimates block importance: XAttention sums anti-diagonals within blocks, FlexPrefill probes attention from the last query block, and MInference applies offline-searched pattern templates per head. Each rests on a *structural prior* about what attention should look like, whereas UniSparse's multi-granularity compression makes no such prior—it computes attention scores directly in a reduced space, inheriting the adaptivity of full attention at a fraction of the cost. When actual attention deviates from these priors—frequent in LongBench v2's reasoning-heavy, length-stratified tasks—structural heuristics misallocate sparsity, while UniSparse's assumption-free, content-driven selection retains genuinely informative blocks, yielding the largest margin on the *Long* subset.

## E. Cross-Hardware Evaluation and End-to-End Latency on H20

While the main paper evaluates efficiency on NVIDIA H200 GPUs across sequence lengths up to 128K, we additionally evaluate on NVIDIA H20 GPUs to verify cross-hardware robustness and extend the analysis to 192K and 256K tokens. We report three complementary measurements: (i) attention-only speedup over FlashAttention, (ii) end-to-end time-to-first-token (TTFT) speedup over FlashAttention (using Meta-Llama-3.1-8B-Instruct), and (iii) block-selection-only speedup of UniSparse over XAttention (the strongest baseline for global block selection). Attention-only and block-selection-only measurements are kernel-level and independent of the model.

Three observations follow from Table 11. First, UniSparse maintains its lead across all sequence lengths on a GPU architecture different from the H200 used in the main paper, confirming that the algorithmic advantage is not tied to a specific hardware platform. Second, the attention-only gap widens as sequence length grows—from $1.62\times$ at 16K to $4.82\times$ at 256K—validating that UniSparse's $\mathcal{O}(L^2 h d_k/(c_q c_k c_h))$ proxy complexity scales favorably as $L$ increases, precisely where the $\mathcal{O}(L^2)$ savings of spar-

*Table 8.* Detailed HELMET task breakdown for **Meta-Llama-3.1-8B-Instruct**. Each section shows performance on six sub-tasks (RAG, ReRank, Cite, LongQA, ICL, Recall) at a specific sequence length. The best results among sparse methods at similar sparsity levels are highlighted.

| Seqlen | Method | RAG | ReRank | Cite | LongQA | ICL | Recall | Overall (↑) |
|---|---|---|---|---|---|---|---|---|
| **8K** | FlashAttention | 70.08 | 65.74 | 32.65 | 26.89 | 71.00 | 100.00 | 61.06 |
| | FlexPrefill (0.95) | 65.62 | 53.43 | 22.61 | 28.45 | 68.52 | 96.69 | 55.89 |
| | XAttention (0.9) | 68.67 | **66.03** | 29.21 | 26.54 | 70.48 | 97.94 | 59.81 |
| | **UniSparse (0.9)** | **69.67** | 64.53 | **30.84** | **29.38** | **70.72** | **100.00** | **60.86** |
| | MInference | **69.67** | 66.59 | 34.37 | 27.08 | 70.92 | **100.00** | **61.44** |
| | FlexPrefill (0.99) | 68.96 | 62.91 | 33.52 | **28.26** | 71.00 | **100.00** | 60.66 |
| | XAttention (0.95) | 69.00 | 66.08 | 30.94 | **28.26** | 71.04 | 99.25 | 60.76 |
| | **UniSparse (0.95)** | 69.58 | 66.25 | 33.84 | 27.40 | **71.16** | **100.00** | 61.37 |
| **16K** | FlashAttention | 67.29 | 51.18 | 24.85 | 33.59 | 75.72 | 99.75 | 58.73 |
| | FlexPrefill (0.95) | 65.08 | 41.86 | 14.58 | **35.16** | 75.44 | 96.25 | 54.73 |
| | XAttention (0.9) | 67.29 | 48.57 | **20.08** | 34.21 | 76.24 | 98.00 | 57.40 |
| | **UniSparse (0.9)** | **67.38** | **49.39** | 18.60 | 34.87 | **76.48** | **99.50** | **57.70** |
| | MInference | 67.33 | 50.28 | 21.17 | 32.74 | 76.24 | **100.00** | 57.96 |
| | FlexPrefill (0.99) | 67.33 | 50.31 | **21.47** | **33.64** | 75.56 | **100.00** | 58.05 |
| | XAttention (0.95) | 67.33 | 50.21 | 20.70 | 33.30 | 76.28 | 99.00 | 57.80 |
| | **UniSparse (0.95)** | **67.50** | **52.32** | 20.78 | 33.45 | **76.52** | 99.50 | **58.34** |
| **32K** | FlashAttention | 65.21 | 43.01 | 10.43 | 42.66 | 78.44 | 99.69 | 56.57 |
| | FlexPrefill (0.95) | 64.58 | 30.38 | 8.35 | 39.28 | **79.16** | 97.06 | 53.14 |
| | XAttention (0.9) | **65.33** | 39.62 | 10.48 | 41.20 | 78.48 | 98.19 | 55.55 |
| | **UniSparse (0.9)** | 65.29 | **40.80** | **10.83** | **41.78** | 79.08 | **99.50** | **56.21** |
| | MInference | 64.92 | 40.31 | **11.59** | 37.65 | **79.36** | 99.12 | 55.49 |
| | FlexPrefill (0.99) | **65.58** | **43.53** | 9.28 | **41.98** | 77.60 | 99.38 | **56.22** |
| | XAttention (0.95) | 65.08 | 40.04 | 10.91 | 41.43 | 78.80 | 99.25 | 55.92 |
| | **UniSparse (0.95)** | 65.12 | 40.80 | 10.22 | 40.70 | 78.84 | **99.50** | 55.87 |
| **64K** | FlashAttention | 64.71 | 33.06 | 11.32 | 45.65 | 81.56 | 99.19 | 55.92 |
| | FlexPrefill (0.95) | 63.58 | 16.22 | 6.15 | 40.36 | 81.92 | 94.44 | 50.45 |
| | XAttention (0.9) | 64.33 | 28.46 | 6.00 | **45.01** | **82.44** | 96.19 | 53.74 |
| | **UniSparse (0.9)** | **64.50** | **29.80** | **6.67** | 44.41 | 82.36 | **97.69** | **54.24** |
| | MInference | 62.21 | 31.02 | 8.52 | 38.59 | **82.88** | 98.94 | 53.69 |
| | FlexPrefill (0.99) | **64.96** | **32.97** | **10.46** | 43.86 | 80.92 | **99.50** | **55.44** |
| | XAttention (0.95) | 64.83 | 29.57 | 9.17 | 44.62 | 82.00 | 98.31 | 54.75 |
| | **UniSparse (0.95)** | 64.58 | 32.41 | 9.50 | **44.81** | 82.04 | 98.31 | 55.27 |
| **128K** | FlashAttention | 58.67 | 14.23 | 2.76 | 44.30 | 83.92 | 94.88 | 49.79 |
| | FlexPrefill (0.95) | **59.88** | 5.19 | 2.54 | 44.32 | 85.40 | **93.19** | 48.42 |
| | XAttention (0.9) | 58.62 | 9.60 | **3.03** | 43.96 | 85.32 | 85.88 | 47.74 |
| | **UniSparse (0.9)** | 59.38 | **9.87** | 2.78 | **44.52** | **85.44** | 91.50 | **48.91** |
| | MInference | 55.88 | **14.35** | 1.72 | 40.58 | 83.84 | 93.38 | 48.29 |
| | FlexPrefill (0.99) | 58.62 | 13.73 | 2.88 | 44.09 | 83.28 | **95.50** | 49.68 |
| | XAttention (0.95) | 59.17 | 12.46 | 2.74 | 45.60 | **84.68** | 91.56 | 49.37 |
| | **UniSparse (0.95)** | **59.21** | 12.76 | **3.23** | **46.54** | 84.36 | 95.25 | **50.22** |

*Table 9.* Detailed HELMET task breakdown for **Qwen2.5-7B-Instruct**. Each section shows performance on six sub-tasks (RAG, ReRank, Cite, LongQA, ICL, Recall) at a specific sequence length.

| Seqlen | Method | RAG | ReRank | Cite | LongQA | ICL | Recall | Overall (↑) |
|---|---|---|---|---|---|---|---|---|
| **8K** | FlashAttention | 60.83 | 58.77 | 27.61 | 34.23 | 69.56 | 79.62 | 55.10 |
| | FlexPrefill (0.95) | 52.33 | 39.94 | 12.69 | 32.32 | 67.92 | 60.19 | 44.23 |
| | XAttention (0.9) | 59.38 | **57.09** | **25.37** | 30.96 | **69.40** | 72.56 | 52.46 |
| | **UniSparse (0.9)** | **59.92** | 56.45 | 24.51 | **33.34** | 69.24 | **75.31** | **53.13** |
| | MInference | **60.79** | **59.04** | 26.07 | 32.28 | **69.92** | 81.38 | **54.91** |
| | FlexPrefill (0.99) | 58.46 | 51.87 | 24.09 | **35.91** | 69.92 | 76.38 | 52.77 |
| | XAttention (0.95) | 59.58 | 57.73 | **27.54** | 32.05 | 69.64 | 77.06 | 53.93 |
| | **UniSparse (0.95)** | 60.21 | 56.57 | 26.36 | 31.75 | 69.52 | 79.06 | 53.91 |
| **16K** | FlashAttention | 59.25 | 37.70 | 16.77 | 37.35 | 74.40 | 77.75 | 50.54 |
| | FlexPrefill (0.95) | 50.67 | 22.92 | 7.83 | 35.25 | 73.32 | 54.75 | 40.79 |
| | XAttention (0.9) | **59.88** | **37.46** | **13.18** | **37.41** | **74.80** | 65.19 | 47.99 |
| | **UniSparse (0.9)** | 58.96 | 37.30 | 13.08 | 37.27 | **74.80** | 70.31 | **48.62** |
| | MInference | 58.38 | 37.54 | 14.87 | 38.69 | 73.84 | **78.88** | **50.36** |
| | FlexPrefill (0.99) | 56.88 | 36.26 | 13.72 | **38.89** | 73.36 | 74.38 | 48.91 |
| | XAttention (0.95) | **60.08** | 38.32 | 15.03 | 36.95 | **74.40** | 72.75 | 49.59 |
| | **UniSparse (0.95)** | 59.42 | **38.48** | **15.46** | 37.34 | 74.36 | 74.69 | 49.96 |
| **32K** | FlashAttention | 56.21 | 25.79 | 13.87 | 42.51 | 76.56 | 76.12 | 48.51 |
| | FlexPrefill (0.95) | 48.75 | 13.43 | 6.73 | 37.44 | **76.52** | 56.38 | 39.87 |
| | XAttention (0.9) | 54.88 | 20.05 | **13.20** | **43.79** | 76.28 | 65.56 | 45.63 |
| | **UniSparse (0.9)** | **55.75** | **25.26** | 11.98 | 41.43 | 76.40 | **70.75** | **46.93** |
| | MInference | 53.96 | 23.96 | 11.13 | 41.55 | 74.16 | **76.81** | 46.93 |
| | FlexPrefill (0.99) | 55.33 | 22.55 | 12.57 | 38.60 | 74.64 | 74.62 | 46.39 |
| | XAttention (0.95) | **55.88** | 25.75 | 13.26 | **42.76** | **76.52** | 68.44 | 47.10 |
| | **UniSparse (0.95)** | 55.71 | **26.26** | **13.59** | 41.90 | 76.40 | 72.69 | **47.76** |
| **64K** | FlashAttention | 55.21 | 14.49 | 10.93 | 44.67 | 77.00 | 66.81 | 44.85 |
| | FlexPrefill (0.95) | 43.92 | 6.83 | 4.07 | 35.17 | 76.56 | 51.12 | 36.28 |
| | XAttention (0.9) | **55.67** | **15.85** | 8.49 | **47.03** | **77.24** | 54.75 | 43.17 |
| | **UniSparse (0.9)** | 55.12 | 13.83 | **10.59** | 46.88 | 77.20 | **61.69** | **44.22** |
| | MInference | 51.83 | 15.46 | 8.54 | 41.85 | 70.56 | **67.50** | 42.62 |
| | FlexPrefill (0.99) | 53.83 | 13.58 | 8.12 | 43.45 | 75.68 | 63.69 | 43.06 |
| | XAttention (0.95) | **55.67** | **15.78** | 11.81 | **47.09** | **77.08** | 60.50 | 44.65 |
| | **UniSparse (0.95)** | 54.96 | 14.25 | **12.45** | 46.94 | 76.88 | 64.19 | **44.95** |
| **128K** | FlashAttention | 46.75 | 7.76 | 6.57 | 45.68 | 79.28 | 49.94 | 39.33 |
| | FlexPrefill (0.95) | 34.12 | 3.75 | 2.48 | 24.07 | 77.48 | 38.44 | 30.06 |
| | XAttention (0.9) | **47.33** | 6.44 | 4.76 | 40.71 | **80.04** | 42.25 | 36.92 |
| | **UniSparse (0.9)** | 47.29 | **7.24** | **5.54** | 40.76 | 80.04 | **47.81** | **38.11** |
| | MInference | 44.54 | 5.34 | 6.29 | 38.15 | 69.28 | 40.69 | 34.05 |
| | FlexPrefill (0.99) | 44.46 | 7.39 | 5.37 | 36.56 | 78.20 | 47.62 | 36.60 |
| | XAttention (0.95) | **47.67** | 8.01 | **6.31** | 42.17 | 79.44 | 45.56 | 38.19 |
| | **UniSparse (0.95)** | 46.88 | **8.34** | 5.36 | **43.19** | 79.88 | 47.81 | **38.58** |

*Table 10.* Comparison on LongBench v2 with Llama-3.1-8B-Instruct. We report accuracy on difficulty (Easy/Hard) and context-length (Short/Medium/Long) subsets, along with overall sparsity and overall score.

| Method | Sparsity ($\uparrow$) | Easy | Hard | Short | Medium | Long | Overall |
|---|---|---|---|---|---|---|---|
| FlashAttention | - | 30.7 | 29.3 | 37.8 | 24.2 | 27.8 | 29.8 |
| XAttention-0.95 | 65.7% | 31.2 | 27.7 | 35.6 | **24.7** | 26.9 | 29.0 |
| FlexPrefill-0.97 | 69.1% | 28.1 | 27.0 | 32.2 | 23.7 | 26.9 | 27.4 |
| MInference | 68.6% | 30.7 | **29.6** | **38.3** | **24.7** | 26.9 | 30.0 |
| **UniSparse-0.95** | 66.7% | **31.7** | **29.6** | 37.8 | **24.7** | **29.6** | **30.4** |

*Table 11.* Speedups on NVIDIA H20 with Meta-Llama-3.1-8B-Instruct (higher is better). Attention-only and TTFT speedups are over FlashAttention; block-selection speedup is over XAttention-0.95.

| Method | 16K | 32K | 64K | 128K | 192K | 256K |
|---|---|---|---|---|---|---|
| *Attention-Only Speedup over FlashAttention* | | | | | | |
| FlexPrefill-0.99 | 1.24× | 1.95× | 2.78× | 3.24× | 3.48× | 3.95× |
| XAttention-0.95 | 1.32× | 1.73× | 2.34× | 2.81× | 3.12× | 3.45× |
| **UniSparse-0.95** | **1.62×** | **2.15×** | **3.03×** | **3.84×** | **4.39×** | **4.82×** |
| *End-to-End TTFT Speedup over FlashAttention* | | | | | | |
| FlexPrefill-0.99 | 1.07× | 1.21× | 1.59× | 2.06× | 2.42× | 2.83× |
| XAttention-0.95 | 1.08× | 1.22× | 1.53× | 1.95× | 2.31× | 2.58× |
| **UniSparse-0.95** | **1.11×** | **1.28×** | **1.70×** | **2.32×** | **2.78×** | **3.24×** |
| *Block-Selection Speedup over XAttention-0.95* | | | | | | |
| **UniSparse-0.95** | **1.16×** | **1.72×** | **2.08×** | **2.30×** | **2.34×** | **2.46×** |

sification compound. Third, end-to-end TTFT speedup is smaller than attention-only speedup (3.24× vs. 4.82× at 256K), reflecting that non-attention components (MLP, normalization) contribute a non-negligible share of total latency; nonetheless, UniSparse attains the highest TTFT speedup among all methods at every sequence length, including a 3.24× end-to-end acceleration at 256K. The block-selection comparison shows an analogous trend: UniSparse's selection cost becomes increasingly lower than XAttention's as $L$ grows (from 1.16× at 16K to 2.46× at 256K), reflecting that multi-granularity compression reduces proxy complexity below XAttention's strided sampling.

## F. Head Compression Analysis

Beyond query-key compression, UniSparse supports optional head-level compression ($c_h > 1$) to further reduce selection overhead. This design leverages the observation that different attention heads often focus on similar important positions, allowing cross-head information to be aggregated without significant accuracy loss. Here we analyze the trade-offs of this optional configuration.

Table 12 evaluates head-level compression under two sparsity settings. Without head compression ($c_h = 1$, our default), UniSparse preserves complete cross-head information and achieves the best accuracy in most cases. Head compression ($c_h = 2$) distills cross-head consensus while reducing selection overhead. However, it reduces achieved sparsity under the same Top-$P$ threshold because aggregating head information makes more blocks appear important

(e.g., on Llama at Top-$P$=0.95, $c_h = 2$ achieves 45.13% vs. 46.33% sparsity).

The efficiency impact reflects this trade-off (§5.3): while head compression dramatically reduces selection overhead (2.64× vs. 1.62× speedup over XAttention), end-to-end speedup is modest (2.61× vs. 2.52× over FlashAttention) due to lower achieved sparsity. Overall, we provide head compression as an optional configuration that is most beneficial when selection overhead dominates or for tasks where attention patterns are more concentrated, with the accuracy-efficiency trade-off being task and model dependent.

*Table 12.* Ablation study on head compression. We evaluate head-level compression factor $c_h$ under two Top-$P$ settings.

| Top-$P$ | Method | Meta-Llama-3.1-8B-Instruct | | | | Qwen2.5-7B-Instruct | | | |
|---|---|---|---|---|---|---|---|---|---|
| | | Sparsity (↑) | RULER (↑) | Sparsity (↑) | HELMET (↑) | Sparsity (↑) | RULER (↑) | Sparsity (↑) | HELMET (↑) |
| 0.9 | **UniSparse** | 58.90% | **92.44** | 61.82% | 55.58 | 60.12% | **83.07** | 60.84% | 46.20 |
| 0.9 | $c_h = 2$ | 57.54% | 92.05 | 60.65% | **55.72** | 53.90% | 83.06 | 54.92% | 46.53 |
| 0.9 | $c_h = 4$ | 56.31% | 91.82 | 59.64% | 55.36 | 46.82% | 82.74 | 47.94% | **46.82** |
| 0.95 | **UniSparse** | 46.33% | **92.56** | 48.65% | **56.21** | 47.25% | 84.26 | 47.51% | 47.03 |
| 0.95 | $c_h = 2$ | 45.13% | 92.37 | 47.73% | 56.02 | 41.73% | **84.27** | 42.27% | **47.27** |
| 0.95 | $c_h = 4$ | 44.45% | 92.30 | 47.39% | 55.93 | 35.32% | 83.99 | 36.18% | 47.10 |

