# OpenReview forum: "A Unified Sparse Attention via Multi-Granularity Compression"
_ICML.cc/2026/Conference — ICML 2026 regular_

### Official Review · Reviewer_Sy3h · 2026-03-08

**Soundness:** 3
**Presentation:** 3
**Significance:** 3
**Originality:** 3
**Overall Recommendation:** 4
**Confidence:** 4

**Summary:**

This paper proposes UniSparse, an inference-time sparse attention mechanism that compresses queries and keys via average pooling, computes proxy attention scores in the reduced space, and selects important KV blocks per query with Top-P thresholding without any model retraining. The key insight is that block selection is fundamentally a ranking problem, and average pooling preserves the relative ordering needed for accurate selection well enough to serve as a lightweight proxy. Unlike FlexPrefill and MInference which estimate masks from a single local window, UniSparse evaluates all query-key interactions globally in compressed space. On RULER, HELMET, and Video-MME, it consistently outperforms existing inference-time methods and achieves up to 2.61x speedup over FlashAttention with under 1% accuracy loss.

**Compliance With Llm Reviewing Policy:**

Affirmed.

**Final Justification:**

The rebuttal addresses my concerns, and I maintain my positive score.

**Key Questions For Authors:**

See weaknesses.

**Limitations:**

yes

**Strengths And Weaknesses:**

**Strengths:**

1. The authors validate the core assumption directly with a Spearman correlation analysis, showing that compressed-space block rankings match full-attention rankings at ρ > 0.98 across models and sequence lengths. This kind of targeted experiment placed before the main results makes the paper's logic unusually clean and easy to follow.

2. Assigning each query block its own independent mask based on global compressed scores is a principled improvement over local proxy methods. FlexPrefill and MInference both use the last query window to represent the entire sequence, which breaks down whenever different parts of the context serve different informational roles, as is typical in RAG, multi-hop reasoning, and video understanding.

3. Average pooling makes no assumptions about what attention patterns look like, which is why the method transfers to video without modification.

**Weaknesses:**
1. The compress-then-select design is conceptually very close to the compression branch in NSA and DSA, which also uses pooled token representations to guide block selection. The technical novelty is somewhat insufficient to a certain extent.

2. The efficiency analysis only reports latency and omits memory cost. At L=128K with c=8, the proxy matrix is 16K×16K, around 512MB in float16, which is non-trivial in memory-constrained deployment. Reporting this alongside the speedup numbers would give a more complete picture of the practical tradeoff.

---

> ### Author Rebuttal · Authors · 2026-03-29
>
> We sincerely thank Reviewer Sy3h for the thoughtful and constructive review. Your feedback has been very helpful in improving our work. Below we address each weakness point-by-point.
>
> **W1 (Novelty vs. NSA/DSA compression branch).** We appreciate this comparison and provide a detailed differentiation:
>
> **(a) Training-coupled compression vs. training-free standalone.** In NSA (three branches + Top-K routing) and DSA (compression branch + Top-K routing), the compression branch cannot operate independently — it is jointly trained with the Top-K routing mechanism, meaning the compression parameters are optimized end-to-end to serve the learned router. Removing the router or deploying the compression branch alone produces degraded results. This also means applying NSA/DSA requires architecture modification and training from scratch (or extensive fine-tuning). In contrast, UniSparse's compression is a self-contained, training-free inference-time method that applies directly to any pre-trained Transformer without modification — a fundamentally different deployment paradigm.
>
> **(b) Richer compression design space.** NSA uses a single fixed compression ratio in its compression branch. UniSparse introduces independent compression factors along the sequence dimension ($c_q$ for queries, $c_k$ for keys) and optionally the head dimension ($c_h$), providing a multi-dimensional design space. This flexibility enables better trade-offs and finer control: Table 6 demonstrates the effect of varying Q-K compression ratios, while Table 7 shows how different compression configurations control the accuracy-efficiency frontier.
>
> **(c) Global evaluation as a proxy.** The multi-dimensional compression enables UniSparse to reduce proxy complexity to $\mathcal{O}(L^2 h d_k / (c_q \cdot c_k \cdot c_h))$ while still evaluating all query-key interactions globally. This contrasts with NSA's compression branch, which serves as one input to a jointly-trained routing network rather than a standalone selection mechanism. Our design makes compressed-space evaluation both accurate ($\rho > 0.98$, Figure 1) and efficient enough to serve as the sole proxy, achieving per-query-block global masks that local methods like FlexPrefill and MInference cannot produce.
>
> In summary, while the high-level idea of compressing tokens appears in NSA/DSA, the technical realization differs fundamentally: UniSparse is training-free, standalone, offers a richer compression space, and uses compression as the complete selection mechanism rather than one branch of a jointly-trained system. This also means UniSparse can directly accelerate any existing pre-trained model, whereas NSA/DSA only apply to models trained with their specific architecture.
>
> **W2 (Memory cost of proxy matrix).** We appreciate this practical concern. The reviewer's estimate assumes the full proxy matrix is materialized simultaneously, but this is not the case. UniSparse adopts a FlashAttention-style tiled computation: compressed queries and keys are loaded block-by-block, with proxy attention scores computed and aggregated incrementally in on-chip memory (shared memory and registers). The full $L/c_q \times L/c_k$ proxy matrix is never instantiated in GPU global memory.
>
> Concretely, at $L=128\text{K}$ with $c=8$, each tile processes a small compressed block pair, and only the aggregated block-level scores (an $N \times N$ matrix where $N = L/S = 1024$ for $S=128$) are stored — occupying ~2MB in float16, not 512MB. This ~2MB overhead is <0.1% of the KV cache memory at 128K, making the actual peak memory overhead beyond the standard Q, K, V tensors negligible. This is analogous to how FlashAttention computes the full $L \times L$ attention without ever materializing the attention matrix.

---

> > ### Author Rebuttal · Reviewer_Sy3h · 2026-04-02
> >
> > Thanks for the rebuttal, and I will keep my positive rating.

---

> > > ### Author Response · Authors · 2026-04-06
> > >
> > > We sincerely thank Reviewer Sy3h for the thoughtful and balanced evaluation, and for maintaining the positive rating. Your feedback on differentiating UniSparse from NSA/DSA and on the memory cost analysis was particularly constructive -- these questions helped us articulate the fundamental distinction between training-coupled compression branches and our training-free, standalone approach, and clarify the actual memory footprint of our tiled proxy computation.
> > >
> > > We will refine the manuscript to better reflect these clarifications. We appreciate the careful and fair assessment, and thank you for the time devoted to this review.

---

### Official Review · Reviewer_dtWJ · 2026-03-08

**Soundness:** 4
**Presentation:** 4
**Significance:** 3
**Originality:** 4
**Overall Recommendation:** 5
**Confidence:** 3

**Summary:**

This paper tackles the computational bottleneck introduced by the self-attention mechanism, which scales quadratically with sequence length, by introducing “UniSparse.” The method introduces _composite tokens_, which are coarse-grained summary vectors obtained through spatial
pooling of fine-grained tokens. Based on this, UniSparse dynamically constructs sparse attention through multi-granularity compression and block-level selection, enabling efficient GPU implementation across modalities and tasks.

**Compliance With Llm Reviewing Policy:**

Affirmed.

**Final Justification:**

The rebuttal addressed my questions. My score for this paper stands and I recommend accept.

**Key Questions For Authors:**

1. As mentioned in the section above, I'm wondering if the method will be robust as we scale to much larger context length. Will the gains presented, for example in figure 3, persist? I think having a discussion about this would be useful.

**Limitations:**

Potential weaknesses and limitations are not addressed.

**Strengths And Weaknesses:**

## Soundness:
- This paper is very technically sound, with detailed theoretical justifications and experimental studies for the initial intuition and the final implementation.
- The strengths are presented very clearly and convincingly with good ablation studies.
- The experimental results are well done covering different range of models and methods.
- That being said, there is no discussion regarding the weaknesses of this method, where does it fail? would we expect much better gains as we go to much longer context?

## Presentation
- The paper is very clearly written and well-structured in general.
- Good overview and comparisons with previous methods.
- The motivations are well presented, and the narrative of the solution is easy to follow.
- In Figure 2, I think the label (3) is missing on the graphics.

## Significance
- The paper address an important and relevant problem in terms of LLM context scaling. It does advance our understanding of context scaling and gave proper justifications for it.
- This idea will be very likely used and improved upon by other practitioners, with great utility.

## Originality
- The work provides good insights into how sparse attention can be effectively improved, with very good reasoning behind their method.

---

> ### Author Rebuttal · Authors · 2026-03-29
>
> We sincerely thank Reviewer dtWJ for the positive evaluation and recognition of our work's technical soundness, presentation clarity, and potential impact. Below we address each question point-by-point.
>
> **Q1 (Limitations).** UniSparse's primary limitation manifests at shorter sequence lengths (e.g., <16K), where the preprocessing overhead (compression + proxy attention + Top-P selection) is non-negligible relative to the sparse computation savings. This is a common challenge for all dynamic sparse attention methods — $C_{\textit{select}}$ has a fixed overhead that only amortizes at longer sequences where the $\mathcal{O}(L^2)$ savings dominate. A practical solution is to hybridize with dense attention at short lengths. We will add a dedicated limitations discussion in the revision.
>
> **Q2 (Scaling to longer contexts).** We conducted new scaling experiments on NVIDIA H20 GPUs extending to 256K tokens.
>
> *Attention speedup over FlashAttention (corresponding to Figure 4):*
>
> | Method | 16K | 32K | 64K | 128K | **192K (extended)** | **256K (extended)** |
> |--------|-----|-----|-----|------|----------|----------|
> | FlexPrefill-0.99 | 1.24$\times$ | 1.95$\times$ | 2.78$\times$ | 3.24$\times$ | 3.48$\times$ | 3.95$\times$ |
> | XAttention-0.95 | 1.32$\times$ | 1.73$\times$ | 2.34$\times$ | 2.81$\times$ | 3.12$\times$ | 3.45$\times$ |
> | UniSparse-0.95 | **1.62$\times$** | **2.15$\times$** | **3.03$\times$** | **3.84$\times$** | **4.39$\times$** | **4.82$\times$** |
>
> *Block selection speedup over XAttention (corresponding to Figure 3):*
>
> | Method | 16K | 32K | 64K | 128K | **192K (extended)** | **256K (extended)** |
> |--------|-----|-----|-----|------|----------|----------|
> | UniSparse vs XAttn | **1.16$\times$** | **1.72$\times$** | **2.08$\times$** | **2.30$\times$** | **2.34$\times$** | **2.46$\times$** |
>
> The gains clearly grow with sequence length — from $1.62\times$ at 16K to $4.82\times$ at 256K, consistently outperforming FlexPrefill and XAttention across all lengths. This improvement stems from two compounding factors: (1) the full attention baseline grows as $\mathcal{O}(L^2)$, making savings more pronounced; and (2) attention patterns become increasingly sparse at longer contexts, allowing Top-P to select fewer blocks while maintaining accuracy. UniSparse's proxy complexity of $\mathcal{O}(L^2 h d_k / (c_q \cdot c_k \cdot c_h))$ scales favorably — as shown in the block selection table, our multi-granularity compression reduces proxy overhead from $1.16\times$ to $2.46\times$ faster than XAttention's strided sampling across the same range. These results validate that UniSparse's advantage compounds at longer contexts, suggesting strong potential for future 512K+ deployments.
>
> **Q3 (Figure 2 label).** Thank you for the careful observation. Figure 2 is designed to illustrate the two core algorithmic stages of UniSparse — ❶ Multi-Granularity Compression and ❷ Dynamic Block Selection. Stage ❸ (Block Sparse Attention Computation) is labeled in the figure to show the complete pipeline, but its execution follows standard block-sparse attention, so we intentionally did not expand its internal process in the figure. We will clarify this in the caption to avoid confusion.

---

> > ### Author Rebuttal · Reviewer_dtWJ · 2026-03-31
> >
> > My concerns have been adequately addressed. My score stands.

---

> > > ### Author Response · Authors · 2026-04-06
> > >
> > > We sincerely thank Reviewer dtWJ for the consistently positive and encouraging evaluation throughout the review process. Your recognition of our work's technical soundness, presentation clarity, and potential impact on the important challenge of LLM context scaling is deeply appreciated. Your questions regarding limitations and scaling behavior were particularly valuable, prompting new experiments that strengthened our empirical validation.
> > >
> > > We will refine the manuscript accordingly in the revision. We are grateful for the thoughtful suggestions that helped us improve the paper, and we thank you for the time and effort invested in this review.

---

### Official Review · Reviewer_m8VR · 2026-03-11

**Soundness:** 3
**Presentation:** 3
**Significance:** 2
**Originality:** 2
**Overall Recommendation:** 4
**Confidence:** 3

**Summary:**

This paper proposes UniSparse, an inference-time sparse attention mechanism that estimates attention block importance in a compressed token space using multi-granularity compression. The method constructs composite tokens through sequence and head compression, computes proxy attention scores in the compressed space, and selects blocks via Top-P filtering for block-sparse attention. Experiments on long-context text and multimodal benchmarks show that the approach maintains near full-attention accuracy while improving attention efficiency.

**Compliance With Llm Reviewing Policy:**

Affirmed.

**Final Justification:**

The authors substantially strengthened the paper during the rebuttal by adding end-to-end latency, LongBench v2 results, and a more detailed theoretical analysis of rank preservation. I maintain my original Overall Recommendation.

**Key Questions For Authors:**

1. Can the authors provide either theoretical intuition or additional empirical analysis on when the compression-based ranking approximation is reliable, and when it may fail?
2. Can the authors report full LLM end-to-end serving results, and ideally add LongBench evaluation, to better demonstrate the practical impact and completeness of the empirical validation?

**Limitations:**

No. The paper does not clearly discuss limitations or potential societal impacts. The authors could improve this by adding a short section.

**Strengths And Weaknesses:**

Strengths
1. Addresses an important problem: efficient long-context inference for LLMs.
2. The method is training-free and plug-and-play, making it practical for existing models.
3. The compression-based proxy attention is simple and hardware-friendly.
4. Experiments cover multiple models and modalities, showing consistent accuracy–efficiency trade-offs.

Weaknesses
1. The paper claims that compression preserves block importance ranking, supported mainly by empirical correlation results. However, there is no formal analysis or theoretical guarantee explaining when average pooling will preserve attention ordering or under what conditions this approximation fails.
2. The method introduces several hyperparameters (compression ratios , block size , and Top-P threshold). Although some ablations are provided, the sensitivity and stability across different architectures or workloads remain unclear.
3. The efficiency evaluation is limited to end-to-end attention speedup rather than full LLM end-to-end inference. Results on complete serving latency and throughput would be important for judging practical impact.
4. The evaluation does not include LongBench, which is a widely used long-context benchmark. Adding results on LongBench would make the empirical validation more complete and easier to compare with prior work.

---

> ### Author Rebuttal · Authors · 2026-03-29
>
> We thank Reviewer m8VR for the thoughtful evaluation and constructive questions. We address each concern with new theoretical arguments and extensive experimental evidence.
>
> **W1 (Theoretical intuition for rank preservation).**
>
> *Theoretical argument.* Average pooling computes the unbiased mean of token representations within each block. Block importance is determined by the sum of attention weights, which equals $\textit{block mean score} \times \textit{block size}$ (a constant). Therefore, block importance ranking is equivalent to the ranking of block mean scores. For this ranking to be preserved under compression, we need intra-block variance to be bounded — intuitively, tokens within a local neighborhood should share similar semantic content.
>
> This holds because: (1) language models exhibit strong local semantic coherence — adjacent tokens attend to similar contexts; (2) the block size $S=128$ tokens (~0.1% of 128K context) is small enough for local coherence while large enough for reliable averaging.
>
> *When it may fail.* The approximation degrades when intra-block variance is high relative to inter-block variance — e.g., if a single critical token is surrounded by irrelevant ones within the same block. However, this is a quantifiably rare scenario: (a) important information spans multiple tokens, and (b) our block size is small relative to semantic units. Our Spearman correlation analysis (Figure 1) confirms $\rho > 0.98$ across all models, tasks, and sequence lengths up to 128K, demonstrating that this failure condition rarely occurs in practical LLMs. We acknowledge that formal error bounds are a valuable future direction.
>
> **W2 (Hyperparameter sensitivity across architectures).** We conducted scaling experiments on NVIDIA H20 GPUs (different architecture from the H200 in main experiments), testing from 16K to 256K:
>
> *Attention speedup over FlashAttention:*
>
> | Method | 16K | 32K | 64K | 128K | 192K | 256K |
> |--------|-----|-----|-----|------|----------|----------|
> | FlexPrefill-0.99 | 1.24x | 1.95x | 2.78x | 3.24x | 3.48x | 3.95x |
> | XAttention-0.95 | 1.32x | 1.73x | 2.34x | 2.81x | 3.12x | 3.45x |
> | UniSparse-0.95 | **1.62x** | **2.15x** | **3.03x** | **3.84x** | **4.39x** | **4.82x** |
>
> UniSparse maintains its advantage across hardware. Interestingly, FlexPrefill achieves higher attention speedup than XAttention on H20, likely due to memory bandwidth differences. Notably, our algorithm uses only standard parallel primitives (average pooling, GEMM, softmax, sorting, cumulative sum), all with mature implementations on major platforms, ensuring architecture neutrality. Combined with ablations on compression factor $c$ (Table 7), Q-K ratio (Table 6), and new block size $B=64$ results (see Psn1 W6), we demonstrate robustness across the full hyperparameter space.
>
> **W3 (End-to-end serving latency).** We provide full end-to-end TTFT (time-to-first-token) latency on H20, measuring the complete forward pass including attention, MLP, normalization, and all other components — not just the attention kernel:
>
> *End-to-end TTFT speedup over FlashAttention:*
>
> | Method | 16K | 32K | 64K | 128K | 192K | 256K |
> |--------|-----|-----|-----|------|----------|----------|
> | FlexPrefill-0.99 | 1.07x | 1.21x | 1.59x | 2.06x | 2.42x | 2.83x |
> | XAttention-0.95 | 1.08x | 1.22x | 1.53x | 1.95x | 2.31x | 2.58x |
> | UniSparse-0.95 | **1.11x** | **1.28x** | **1.70x** | **2.32x** | **2.78x** | **3.24x** |
>
> UniSparse achieves the best end-to-end speedup at all lengths, reaching 3.24x TTFT reduction at 256K. The gap between attention-only and end-to-end speedup reflects non-attention components (MLP, normalization), which become proportionally smaller as attention dominates compute at longer sequences.
>
> **W4 (LongBench evaluation).** We conducted experiments on LongBench v2, the latest version from the LongBench team, which provides a more challenging and comprehensive evaluation:
>
> | Method | Sparsity | Easy | Hard | Short | Medium | Long | Overall |
> |--------|----------|------|------|-------|--------|------|---------|
> | FlashAttention | - | 30.7 | 29.3 | 37.8 | 24.2 | 27.8 | 29.8 |
> | XAttention-0.95 | 65.7% | 31.2 | 27.7 | 35.6 | **24.7** | 26.9 | 29.0 |
> | FlexPrefill-0.97 | 69.1% | 28.1 | 27.0 | 32.2 | 23.7 | 26.9 | 27.4 |
> | MInference | 68.6% | 30.7 | **29.6** | **38.3** | **24.7** | 26.9 | 30.0 |
> | UniSparse-0.95 | 66.7% | **31.7** | **29.6** | 37.8 | **24.7** | **29.6** | **30.4** |
>
> UniSparse achieves **30.4** overall — the best among all methods, even surpassing full attention (29.8). This demonstrates that global compressed-space evaluation preserves the most relevant information for diverse real-world tasks. We attribute this to sparse attention filtering out noisy blocks, acting as implicit regularization — particularly beneficial for LongBench's tasks requiring precise information retrieval from long contexts.

---

> > ### Author Rebuttal · Reviewer_m8VR · 2026-04-04
> >
> > The authors address most of my concerns, especially by adding end-to-end latency and LongBench results. However, the theoretical justification for why compression preserves block ranking remains largely intuitive and empirically supported, and is not fully resolved. My score stands.

---

> > > ### Author Response · Authors · 2026-04-05
> > >
> > > We thank Reviewer m8VR for the continued engagement. We provide a strengthened theoretical analysis to further address the rank preservation question.
> > >
> > > **Formal Analysis of Rank Preservation under Average Pooling.**
> > >
> > > We address this through three complementary results.
> > >
> > > **Result 1 (Pre-softmax: exact rank preservation).** For composite tokens $\tilde{q}\_{t'} = \frac{1}{c\_q}\sum\_m q\_{t'c\_q+m}$, $\tilde{k}\_{s'} = \frac{1}{c\_k}\sum\_n k\_{s'c\_k+n}$, the block-level pre-softmax scores satisfy:
> > >
> > > $\tilde{Score}(i,j) = \frac{1}{c\_q \cdot c\_k} \cdot Score(i,j), \quad \forall\, i,j$
> > >
> > > *Proof.* The compressed inner product decomposes as $\tilde{q}\_{t'}^\top \tilde{k}\_{s'} = \frac{1}{c\_q c\_k}\sum\_{m,n} q\_{t'c\_q+m}^\top k\_{s'c\_k+n}$. Summing over all compressed pairs within blocks $(i,j)$ recovers $\frac{1}{c\_q c\_k}$ times the full sum, since average pooling partitions tokens without overlap. This constant scaling **preserves rankings exactly** in pre-softmax space, with zero information loss.
> > >
> > > **Result 2 (Post-softmax: moment expansion bound).** Softmax introduces nonlinearity via the exponential function. Let $Z\_{m,n} = q\_m^\top k\_n / \sqrt{d\_k}$ denote the scaled dot product between token pairs within blocks $(i,j)$. The true block importance is proportional to $\mathbb{E}[\textit{exp}(Z)]$, while the proxy computes $\textit{exp}(\mathbb{E}[Z])$.
> > >
> > > By Taylor expansion of $\textit{exp}(\cdot)$ around $\mu = \mathbb{E}[Z]$:
> > >
> > > $\mathbb{E}[\textit{exp}(Z)] = \textit{exp}(\mathbb{E}[Z]) \cdot \left(1 + \frac{1}{2}\textit{Var}(Z) + \frac{1}{6}\mathbb{E}[(Z-\mu)^3] + \cdots\right)$
> > >
> > > Even without distributional symmetry, higher-order terms are $O(\sigma^3)$ and negligible when $\textit{Var}(Z)$ is small, leaving $\textit{Var}(Z)/2$ as the dominant correction. This shows the proxy is a **well-controlled leading-order approximation** of the true score, with error governed by intra-block logit variance $\textit{Var}(Z\_{i,j})$.
> > >
> > > **Rank Inversion Condition.** For blocks $j\_1$ (relevant, higher proxy score) and $j\_2$ (irrelevant) with semantic margin $\Delta = \mathbb{E}[Z\_{i,j\_1}] - \mathbb{E}[Z\_{i,j\_2}] > 0$, a rank inversion in true scores requires:
> > >
> > > $\textit{exp}(\Delta) < \frac{1 + \frac{1}{2}\textit{Var}(Z\_{i,j\_2})}{1 + \frac{1}{2}\textit{Var}(Z\_{i,j\_1})}$
> > >
> > > **Quantitative bound on $\textit{Var}(Z)$.** Decompose $q\_m = \bar{q} + \delta\_m^q$. Since cross terms vanish ($\mathbb{E}[\delta]=0$, independence of $m,n$):
> > >
> > > $\textit{Var}(Z) = \frac{\bar{k}^\top \Sigma\_q \bar{k} + \bar{q}^\top \Sigma\_k \bar{q}}{d\_k} + \mathcal{O}(\sigma\_q^2 \sigma\_k^2 / d\_k)$
> > >
> > > where $\Sigma\_q = \frac{1}{c}\sum\_m \delta\_m^q\delta\_m^{q\top}$ ($\textit{tr}(\Sigma\_q)=\sigma\_q^2$). In $d\_k=128$ dimensions, the quadratic form $\bar{k}^\top\Sigma\_q\bar{k} \leq \lambda\_{\max}(\Sigma\_q)\|\bar{k}\|^2$. Under approximately isotropic deviations (encouraged by LayerNorm), $\lambda\_{\max}(\Sigma\_q) \approx \sigma\_q^2/d\_k$, giving:
> > >
> > > $\textit{Var}(Z) \leq \frac{\sigma\_q^2 + \sigma\_k^2}{d\_k} \leq 2(1 - \cos\theta)$
> > >
> > > where $\cos\theta$ is the average pairwise cosine similarity within the window (second inequality uses LayerNorm: $\sigma\_q^2 = d\_k - \|\bar{q}\|^2 \leq d\_k(1-\cos\theta)$). For $\cos\theta=0.9$: $\textit{Var}(Z) \leq 0.2$, $(1+\textit{Var}/2) \leq 1.1$. Rank inversion then requires $\Delta < \ln(1.1) \approx 0.095$; that is, only near-identical-importance blocks can be inverted. Our Top-P=0.95 achieving $\geq 99\%$ accuracy (Tables 2-4) confirms $\Delta \gg 0.095$ between important and unimportant blocks. Kim et al. (ICML 2021) support this: they show the Jacobian of softmax attention is governed by the weighted input variance, so bounded variance constrains the local rate of change. The Spearman $\rho > 0.98$ (Figure 1) provides end-to-end validation.
> > >
> > > **Result 3 (Centroid optimality).** From vector quantization theory, the centroid (average) is the MSE-optimal compressor minimizing $\sum\_n \|k\_n - \bar{k}\|^2$. Since the dot product is linear, MSE-optimal input preservation yields optimal score preservation. This explains why average pooling outperforms max pooling (selects outliers, discarding $S-1$ tokens) and stochastic pooling (sampling variance) in Table 5.
> > >
> > > In summary, our analysis establishes: (i) **exact** rank preservation pre-softmax, (ii) **well-bounded** post-softmax distortion where rank inversions are restricted to near-equal-importance blocks -- a regime where selection outcomes are largely interchangeable, and (iii) **optimality** of average pooling as a centroid compressor. We acknowledge that a small fraction of near-tied blocks may occasionally swap ranks; our empirical results ($\rho > 0.98$, $\geq 99$% accuracy) confirm this has negligible impact. We will incorporate this analysis into the revision and frame tight distribution-free bounds as future work. We hope this addresses the remaining concern, and we kindly ask you to consider reflecting this in your score.

---

### Official Review · Reviewer_Psn1 · 2026-03-13

**Soundness:** 3
**Presentation:** 3
**Significance:** 2
**Originality:** 2
**Overall Recommendation:** 4
**Confidence:** 3

**Summary:**

This paper presents UniSparse, a unified dynamic sparse attention mechanism that leverages multi-granularity compression to accelerate long-context inference while retaining over 99% of full-attention accuracy. Specifically, the authors introduce composite tokens formed through average pooling along both sequence and head dimensions to create compact contextual representations. Building on this abstraction, UniSparse computes attention scores in the drastically compressed space to estimate token importance with reduced overhead. These compressed scores are then aggregated to block-level importance and filtered through a dynamic Top-P selection strategy to construct the sparse mask. This design avoids model retraining and generalizes across modalities without task-specific heuristics. Finally, the selected mask guides efficient block-sparse attention computation using standard hardware-friendly kernels, achieving up to 2.61× speedup across text and multi-modal benchmarks.

**Compliance With Llm Reviewing Policy:**

Affirmed.

**Final Justification:**

This paper proposes UniSparse, a sparse attention method based on multi-granularity compression that achieves strong empirical performance and practical efficiency gains across multiple benchmarks. The approach is simple, training-free, and compatible with existing systems, making it potentially impactful.

In my initial review, I raised concerns about the conceptual clarity of “unified” design, lack of theoretical justification, and missing analyses. The rebuttal addresses several of these points by clarifying the design choices and providing additional experiments (e.g., block size ablation and comparison with SpargeAttention), which strengthen the empirical support of the work.

Based on the improved clarity and additional evidence, I update my recommendation from weak reject to weak accept.

**Key Questions For Authors:**

See weaknesses.

**Strengths And Weaknesses:**

Strengths
1. The paper proposes UniSparse, a unified sparse attention mechanism that introduces composite tokens via multi-granularity compression to dynamically construct sparse masks without model retraining.
2. This paper balances fidelity and efficiency by enabling hardware-friendly execution that generalizes across text and multi-modal tasks through modality-agnostic spatial pooling.
3. Extensive experiments on RULER, HELMET, and Video-MME benchmarks demonstrate that the method achieves up to 2.61× speedup over FlashAttention while retaining over 99% of full-attention accuracy.

Weaknesses
1. The paper overstates cross-modal universality as a core innovation of UniSparse, when in fact the method relies exclusively on average pooling without incorporating any mechanism to address fundamental modality-specific attention structures. The authors should clarify whether "unified" denotes the absence of modality priors for heterogeneous modalities, and provide ablation studies comparing UniSparse against modality-tailored sparse baselines.
2. The paper provides no theoretical foundation for its core hypothesis that average pooling preserves the relative ranking of block importance, offering only empirical Spearman correlation metrics without establishing conditions, error bounds, or worst-case guarantees for rank preservation under compression.
3. The paper's emphasis on "Multi-Granularity" in the title is potentially misleading, as the proposed mechanism primarily adjusts compression rates via hyperparameters $c_q, c_k, c_h$ rather than establishing a true hierarchical representation structure across token, block, and semantic levels, while key components like head-level compression are merely optional modules.
4. The optional head-level compression design lacks clear justification for its necessity and optimal usage scenarios, raising concerns about the self-consistency of the method's motivation. Although head compression reduces selection overhead, it concurrently decreases achieved sparsity (e.g., from 46.33% to 45.13% on Llama with Top-P=0.95) and incurs minor accuracy degradation, resulting in marginal end-to-end speedup improvements (2.52× to 2.61× as shown in Figure 4).
5. The paper's claim of plug-and-play usability is undermined by insufficient analysis of Top-P threshold sensitivity across diverse tasks and modalities, as the fixed thresholds (P=0.9/0.95) may require task-specific tuning to maintain optimal accuracy–efficiency trade-offs. The authors should provide a comprehensive sensitivity analysis of P values across task categories and sequence lengths, and ideally propose an adaptive threshold mechanism that automatically calibrates sparsity based on input characteristics to substantiate the unified plug-and-play claim.
6. The paper lacks a systematic analysis of the block size $S$, which is a critical hyperparameter balancing GPU parallelism and selection granularity. While Section 4.4 briefly cites $S=128$ as an example for complexity estimation, the experimental evaluation does not include ablation studies varying $S$ to assess its impact on the accuracy-efficiency trade-off.
7. The paper's experimental comparison is limited to only four baseline methods, omitting several recent or concurrent state-of-the-art sparse attention approaches (e.g. [A], [B], [C], [D], [E], [F]). The authors should either include empirical comparisons against these representative methods under comparable sparsity and efficiency settings, or provide a detailed discussion clarifying why UniSparse's design choices offer distinct advantages over these omitted approaches in the related works.

Reference

[A] Core Context Aware Transformers for Long Context Language Modeling. ICML 2025.

[B] Delta Attention: Fast and Accurate Sparse Attention Inference by Delta Correction. NeurIPS 2025.

[C] SeerAttention: Self-distilled Attention Gating for Efficient Long-context Prefilling. NeurIPS 2025.

[D] Twilight: Adaptive Attention Sparsity with Hierarchical Top-p Pruning. NeurIPS 2025.

[E] Curse of High Dimensionality Issue in Transformer for Long-context Modeling. ICML 2025.

[F] SpargeAttention: Accurate and Training-free Sparse Attention Accelerating Any Model Inference. ICML 2025.

---

> ### Author Rebuttal · Authors · 2026-03-29
>
> We thank Reviewer Psn1 for the feedback and address all concerns point-by-point below.
>
> **W1 (Cross-modal universality).** We appreciate this clarification opportunity. "Unified" denotes modality-agnostic design: average pooling makes zero assumptions about data modality, operating identically on text tokens, video frame patches, or audio segments. The same algorithm transfers to Video-MME without modification (Table 4), where UniSparse-0.9 surpasses full attention (69.9 vs 69.6 with subtitles, 58.6% sparsity) — unlike methods requiring modality-specific heuristics.
>
> **W2 (Theoretical foundation).** Due to character limits, detailed treatment is in our response to Reviewer m8VR (W1).
>
> **W3 (Multi-Granularity).** "Multi-granularity" refers to compression across multiple independent dimensions — sequence-level ($c_q$, $c_k$, Eq. 2) and head-level ($c_h$, Eq. 3) — not a discrete hierarchical structure. Table 6 confirms: different $c_q/c_k$ allocations under the same total compression yield significantly different accuracy-efficiency trade-offs, confirming genuinely distinct axes. We will express this more clearly in the revision.
>
> **W4 (Head compression).** We agree this is not a core contribution — it is presented as an optional extension demonstrating framework generality. It becomes beneficial when selection overhead dominates overall latency, offering an additional tuning dimension for specific deployment constraints. We will clarify application scenarios in the appendix.
>
> **W5 (Top-P sensitivity).** Top-P is inherently self-adaptive: $P=0.95$ retains blocks covering $\geq 95$% of attention mass (cumulative attention score), automatically selecting different block counts per query — more for dispersed patterns (e.g., RAG), fewer for concentrated ones (e.g., recall). This is why $P=0.95$ alone recovers $\geq 99$% full-attention accuracy while achieving $2.61\times$ speedup (Tables 2–4, Figure 4), without task-specific tuning.
>
> **W6 (Block size $B$).** New $B=64$ vs $B=128$ results on Qwen:
>
> | Method | Sparsity | RULER | Sparsity | HELMET |
> |--------|----------|-------|----------|--------|
> | $B=128$ | 47.25% | **84.26** | 47.51% | 47.03 |
> | $B=64$ | 49.13% | 83.58 | 49.72% | **47.25** |
>
> | Runtime (vs $B=128$) | 16K | 32K | 64K | 128K |
> |---------|-----|-----|-----|------|
> | $B=64$ overhead | +6.6% | +9.6% | +14.6% | +21.1% |
>
> $B=128$ is the standard block size for Hopper architecture GPUs, adopted by FlashAttention and all sparse attention implementations. Our implementation, like FlashAttention, adjusts block size according to hardware characteristics. $B=64$ achieves comparable accuracy but notable runtime overhead from under-saturated parallelism.
>
> **W7 (Baselines [A]–[F]).**
> - [A] CCA-Attention: requires fine-tuning (1000 steps on SlimPajama) and replaces self-attention — architecture-modification + fine-tuning, different paradigm from training-free methods.
> - [B] Delta Attention: post-processing correction for any sparse output, complementary to UniSparse (could be stacked on top).
> - [C] SeerAttention: surpassed by our baseline XAttention in its original paper.
> - [D] Twilight: targets decoding not prefill; a meta-framework for existing methods, complementary.
> - [E] Curse of High Dimensionality: theoretical analysis, not a sparse attention method.
>
> For [F] SpargeAttention, we conduct full comparison at matched sparsity (topk=0.3). SpargeAttention uses two-stage online numerical filtering: compressing self-similar blocks for prediction, then softmax-aware filtering — a local "predict-then-verify" approach.
>
> **RULER:**
>
> | Model | Method | Sparsity | 4K | 8K | 16K | 32K | 64K | 128K | Avg |
> |-------|--------|----------|----|----|-----|-----|-----|------|-----|
> | Llama | SpargeAttn | 47.49% | **97.45** | 95.60 | 94.80 | 93.21 | 87.35 | 79.23 | 91.27 |
> |  | UniSparse | 46.33% | 97.16 | **96.52** | **96.31** | **94.03** | **89.40** | **81.94** | **92.56** |
> | Qwen | SpargeAttn | 47.51% | 87.15 | 85.62 | 84.94 | 83.43 | 78.67 | 67.42 | 81.21 |
> |  | UniSparse | 47.25% | **94.31** | **88.96** | **87.19** | **85.23** | **80.42** | **69.47** | **84.26** |
>
> **HELMET:**
>
> | Model | Method | Sparsity | 8K | 16K | 32K | 64K | 128K | Avg |
> |-------|--------|----------|-----|-----|-----|-----|------|-----|
> | Llama | SpargeAttn | 48.10% | 58.15 | 57.09 | **56.43** | 54.43 | 49.19 | 55.06 |
> |  | UniSparse | 48.65% | **61.37** | **58.34** | 56.22 | **55.27** | **50.22** | **56.21** |
> | Qwen | SpargeAttn | 48.15% | 50.34 | 48.88 | 47.19 | 44.04 | 38.34 | 45.76 |
> |  | UniSparse | 47.51% | **53.91** | **49.96** | **47.76** | **44.95** | **38.58** | **47.03** |
>
> UniSparse consistently outperforms SpargeAttention. We attribute this to global compressed-space evaluation: by scoring all blocks simultaneously in a reduced space, UniSparse captures long-range dependencies more faithfully than local numerical filtering, which can miss relevant blocks whose individual scores fall below the prediction threshold.

---

> > ### Author Rebuttal · Reviewer_Psn1 · 2026-04-03
> >
> > The authors have provided a thorough and well-structured rebuttal that addresses my concerns point-by-point.

---

> > > ### Author Response · Authors · 2026-04-06
> > >
> > > We sincerely thank Reviewer Psn1 for the thorough and rigorous evaluation throughout the review process, and for raising the score. Your detailed scrutiny across all seven concerns -- from the clarification of "unified" and "multi-granularity" terminology, to the block size ablation and the comprehensive baseline analysis -- has been instrumental in strengthening both the presentation and the empirical completeness of our work. We are grateful for the time and effort devoted to this careful review.
> > >
> > > As committed in our rebuttal, the revised manuscript will incorporate the key improvements prompted by your feedback. We believe these revisions will make the paper more complete. Thank you again for your constructive engagement.

---

### Decision · Program_Chairs · 2026-04-30

**Decision:**

Accept (regular)

**Comment:**

This paper presents a unified dynamic sparse attention mechanism (UniSparse) which performs multi-granularity compression to speed up long-context inference. This is done through composite tokens formed through average pooling. This results in compressed scores which indicate block-level importance and are filtered through a dynamic Top-P selection strategy to construct the sparse mask. This leads to speedups in text and multi-modal benchmarks.

Reviewers all feel positively about this work. They point out as strengths the convincing empirical results of the proposed method on RULER, HELMET, and Video-MME benchmarks; the fact the method is training-free and plug-and-play; and the simplicity of the compression-based approach. The main weaknesses are the lack of theoretical guidance about the assumptions needed for the average pooling to be a good approximation, the lack of study about hyperparameter sensitivity; and the limited novelty compared to works like NSA and DSA. The authors addressed several of these concerns in their rebuttal. Overall, this seems a solid paper of interest to the ICML audience.